# Kolmogorov-Arnold Fourier Networks

## Abstract

Although Kolmogorov-Arnold Networks (KAN) based on the Kolmogorov-Arnold theorem (Kolmogorov, 1933) possess strong theoretical expressiveness, they face severe scalability bottlenecks—specifically parameter explosion and difficulty in capturing high-frequency features—in high-dimensional tasks. To address these issues, we propose the **Kolmogorov-Arnold-Fourier Network (KAF)**, which fundamentally redefines the KAN paradigm through spectral reparameterization. Our key contributions include: (1) proposing a fundamental **basis transformation** from the local, grid-based B-spline representation to a global, adaptive spectral representation. This shift changes the network's inductive bias, reducing parameter complexity from $O(G)$ to $O(1)$ while preserving expressiveness; (2) introducing **trainable** Random Fourier Features (RFF (Tancik et al., 2020; Bracewell, 1986)) initialized via a spectral alignment strategy, which allows the model to break the smoothness limitation of fixed kernels and accurately capture high-frequency components; and (3) implementing an adaptive hybrid GELU-Fourier activation mechanism that progressively enhances frequency representation during training. Comprehensive experiments demonstrate the superiority of KAF across vision, NLP, audio, and PDE solving tasks, achieving state-of-the-art performance (e.g., 93.15% on CIFAR-10) with significantly improved efficiency. We will release the source code in accordance with the review policy.

## 1 Introduction

The interpretability of deep neural networks (Howard et al., 2017; Han et al., 2016) has long been one of the core challenges in the field of machine learning. The Kolmogorov-Arnold(Liu et al., 2024; Kolmogorov, 1933) theorem states that any continuous multivariate function can be represented through a combination of univariate functions (Mhaskar & Micchelli, 1996; Barron, 1993). This theory provides significant inspiration for the design of interpretable neural network architectures. Based on this theory, Kolmogorov-Arnold Networks (KAN) (Liu et al., 2024; Schmidt-Hieber, 2021) have been proposed, which replace the fixed activation functions in traditional multilayer perceptrons (MLPs (Rumelhart et al., 1986)) with learnable B-spline (De Boor, 1972) basis functions, theoretically demonstrating strong expressive potential and flexibility. By introducing trainable nonlinear activation functions, KAN enables the network to dynamically adjust the shape of the activation functions according to the characteristics of the data, thereby enhancing the adaptability and performance of the model.

However, despite the significant theoretical advantages of KAN, its practical application faces two fundamental issues that severely limit its generalization and adoption in high-dimensional tasks: Inefficient Parameter Utilization: The dual-matrix architecture of KAN (i.e., the activation function matrix and the B-spline coefficient matrix) leads to a rapid increase in the number of parameters. Compared to traditional MLPs, where the parameter count scales with input × output + bias, KAN's parameter count grows several times larger. This makes it challenging to apply KAN to high-dimensional tasks such as computer vision. The explosion in parameters (Xu et al., 2019; Mhaskar & Micchelli, 1996;

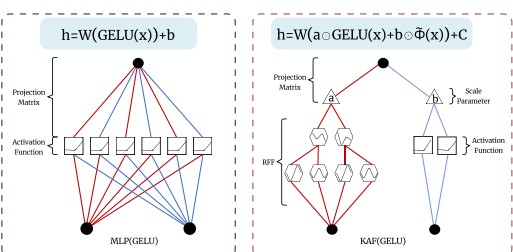

Figure 1: Comparison between a standard GELU-MLP and a GELU-KAF.

Barron, 1993) not only increases the storage and computational costs of the model but also significantly reduces the efficiency of both training and inference (Tan & Le, 2020; 2021).The B-spline (De Boor, 1972; Eldar & Unser, 2021; Aziznejad et al., 2023; Li et al., 2023; Wu et al., 2022)basis functions employed by KAN exhibit inherent spectral limitations when performing function approximation in high-dimensional spaces. The smoothness of B-spline basis functions makes it difficult to accurately capture high-frequency components of signals, leading to suboptimal performance when processing data with rich spectral features, such as natural images or audio waveforms. This limitation in spectral representation capability adversely affects the model's performance and stability in practical applications(Xu et al., 2019). This dilemma creates a paradox between theory and practice: although KAN theoretically encompasses all functionalities of MLPs, its inefficiency and spectral distortion issues force practitioners to make a trade-off between interpretability and scalability.

To address the aforementioned issues, the key challenge is balancing the inherent trade-off between model interpretability and parameter efficiency, which has long plagued traditional neural networks. This paper make an attempt to fundamentally redefine the traditional KAN paradigm (Bracewell, 1986). We attempt to address the bottlenecks of the Kolmogorov-Arnold theory in practical applications (i.e., the parameter explosion and insufficient high-frequency capture problems of the KAN model), so that it can be applied to high-dimensional, large-scale modern AI tasks. Specifically, this paper introduces an innovative neural network architecture—**Kolmogorov-Arnold-Fourier Networks (KAF)**, which employs Fourier domain reparameterization and dynamic activation evolution, aiming to bridge the gap between interpretability and parameter efficiency. Our main contributions include: (1) proposing a fundamental **basis transformation** from the local, grid-based B-spline representation (KAN) to a global, adaptive spectral representation. This shift changes the network's inductive bias, breaking the smoothness limitation of fixed kernels and enabling efficient high-dimensional scaling. Consequently, we resolve the parameter explosion issue, reducing complexity from $O(d_{in} \times d_{out} \times (G + K + 3))$ to $O(d_{in} \times d_{out})$ (See Supp.3.3 and Supp.A for proof) while preserving expressiveness; (2) We replace the traditional B-spline basis functions with trainable **Random Fourier Features**. We replace the traditional B-spline basis functions with trainable **Random Fourier Features** to eliminate the need for the spline coefficient matrix. An initialization strategy based on the Central Limit Theorem ($\sigma = 1.64$) aligns the RFF spectrum with the prior knowledge of natural signals, avoiding spectral leakage issues. This significantly enhances the model's spectral fidelity and expressive power in high-dimensional spaces.

We design a hybrid GELU-Fourier activation function with learnable coefficients $\{a, b\}$. During training, these coefficients are dynamically adjusted through gradient backpropagation, enabling an automatic transition from fixed activation functions to hybrid Fourier-symbolic representations.

## 2 RELATED WORK

**Multi-Layer Perceptrons and Current Challenges.** The design and optimization of deep learning (Touvron et al., 2021)models remain central to machine learning research. Traditional MLPs(Rumelhart et al., 1986), among the earliest neural networks (Han et al., 2016), offer simplicity and scalability, with rich theoretical foundations. While ResNet (He et al., 2015b) and Transformer (Vaswani et al., 2023) models have shown remarkable performance across various tasks, MLPs face challenges in theoretical interpretability and practical bottlenecks. Traditional activation functions like ReLU (Nair & Hinton, 2010; Glorot et al., 2011) and Sigmoid (Elfwing et al., 2017) often fail to adapt to complex data, and despite their efficiency, MLPs struggle with high-frequency features and complex distributions. Improving activation mechanisms and parameter efficiency has become crucial for enhancing MLPs' adaptability to high-dimensional data.

**Kolmogorov-Arnold Networks and Scalability Issues.** The Kolmogorov-Arnold (Fan et al., 2019) Theorem underpins networks for approximating continuous multivariable functions. The pioneering KAN replaced fixed activations with B-spline (De Boor, 1972) functions but faces challenges in high-dimensional applications due to parameter explosion and GPU inefficiency. Recent improvements include KAN-Transformer, MLP-KAN with sparse parameters, and FAN(Dong et al., 2024) with Fourier activations, all seeking to balance interpretability with scalability. Also, Previous work (Xu et al., 2024; Mehrabian et al., 2025; Ai et al., 2025; Rong et al., 2025; Zhou et al., 2024) has explored the combination of Fourier and KAN architectures in different scenarios.

**Enhancing Spectral Representation with KAF.** To address high-frequency modeling challenges, Random Fourier Features (RFF (Rahimi & Recht, 2007b; Yu et al., 2016)) enable spectral domain mapping, with variants like Learnable RFF and SIREN enhancing expressiveness. Our proposed KAF incorporates GELU and learnable Fourier features, with scale factor control and variance initialization. This reduces parameters while improving spectral representation. KAF maintains KAN's interpretability while enhancing scalability and efficiency, showing superior performance in capturing high-frequency (Sitzmann et al., 2020) details across NLP, vision, audio, and traditional machine learning tasks.

# 3 METHODOLOGY

## 3.1 KOLMOGOROV-ARNOLD THEOREM

The Kolmogorov-Arnold (Bracewell, 1986; Kolmogorov, 1933) theorem, proposed by Soviet mathematicians Vladimir Arnold and Andrey Kolmogorov in the 1950s, states that any continuous (Maiorov & Pinkus, 2021) multivariate function $f : [0,1]^d \rightarrow \mathbb{R}$ can be represented as a superposition of univariate functions:

$$f(x_1, x_2, \ldots, x_d) = \sum_{q=1}^{2d+1} \Phi_q \left( \sum_{p=1}^{d} \phi_{q,p}(x_p) \right),$$

where $\Phi_q : \mathbb{R} \rightarrow \mathbb{R}$ and $\phi_{q,p} : [0,1] \rightarrow \mathbb{R}$ are univariate continuous functions. This theorem provides a theoretical foundation for dimensionality reduction in high-dimensional function approximation.

In (Berner et al., 2022b; Poggio et al., 2017), the theorem suggests that high-dimensional functions can be captured through low-dimensional (Xu et al., 2019) transformations, resembling the hierarchical structure of neural networks.

## 3.2 KOLMOGOROV-ARNOLD NETWORK (KAN)

Although the Kolmogorov-Arnold (Bracewell, 1986)theorem was proposed quite early, (KAN (Liu et al., 2024)) is proposed according to this theorem, demonstrating that this structure can, in a sense, serve as an alternative to traditional MLP models. In the KAN network, each layer can be represented by the following formula:

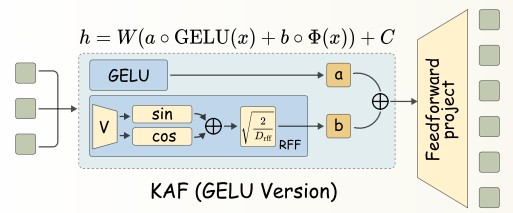

Figure 2: Pipeline of the KAF (GELU Version) layer, where input is processed through parallel GELU and Random Fourier Feature (RFF) branches, scaled by 'a' and 'b' respectively, summed, and then passed to a final linear projection.

$$f(\mathbf{x}) = \Phi \circ \mathbf{x} = \left[ \sum_{i=1}^{d_{in}} \phi_{1,i}(x_i) \quad \cdots \quad \sum_{i=1}^{d_{in}} \phi_{d_{out},i}(x_i) \right],$$

where $\Phi$ is a matrix of basis functions. This formula aligns with the form of the Kolmogorov-Arnold theorem. However, in practical applications, they chose B-spline (Bach, 2016) basis functions as the basis functions $\phi_{q,p}$, and added an external activation function $SILU$ to guide the update of the KAN layer(Ramachandran et al., 2017; Elfwing et al., 2017). The formula can be expressed as

$$\phi(x) = w_h \text{silu}(x) + w_s \text{spline}(x),$$
$$\text{spline}(x) = \sum_i c_i B_i(x).$$

Among them, $\Phi$ represents the basis function matrix, where B-spline basis functions and the SiLU activation function were used. However, KAN suffers from excessive parameter growth, with a parameter count of $d_{\text{out}} \times d_{\text{out}} \times (G + K + 3) + d_{\text{out}}$, far exceeding MLP's $d_{\text{in}} \times d_{\text{out}} + d_{\text{out}}$, while

also being computationally inefficient on GPUs and failing to capture high-frequency components, limiting its practical applicability.

### 3.3 KAF: KOLMOGOROV-ARNOLD FOURIER NETWORK

In the previous discussion, we pointed out that traditional networks based on the Kolmogorov-Arnold theorem (KAN) often face multiple challenges in practical applications. To address these issues, we propose an alternative approach—KAF (Kolmogorov-Arnold Fourier Network). By replacing B-spline basis functions with Random Fourier Features (RFF(Fathony et al., 2021; Tancik et al., 2020; Bracewell, 1986)), which are more efficient for GPU acceleration, and introducing hybrid spectral correction for the activation functions, the network retains the advantages of the Kolmogorov-Arnold theory while achieving training efficiency and inference speed closer to that of MLPs (Rumelhart et al., 1986). This section provides a detailed explanation of the overall architecture of the KAF network, the utilization of Random Fourier Features within the network, the design and scaling principles of the GELU-Fourier hybrid activation function, as well as the RFF weight initialization strategy and the theoretical justification for $\sigma = 1.64$.

**Overall Architecture.** In the overall framework of KAF, we follow the core idea of the Kolmogorov-Arnold theorem, which approximates high-dimensional target functions through the composition of several low-dimensional learnable functions. Unlike the KAN network, which directly utilizes B-spline basis functions, KAF employs Random Fourier Features (RFF) in each layer to perform a nonlinear mapping of the input, and then uses linear transformations to achieve the composition of the "outer function" and the "inner function." Specifically, the core computational process of each KAF layer can be formulated as:

$$\mathbf{h}^{(l)} = \underbrace{\mathbf{W}^{(l)}}_{\text{outer function}} \left( \underbrace{\mathbf{a}^{(l)} \odot \text{GELU}(\tilde{\mathbf{x}}^{(l)}) + \mathbf{b}^{(l)} \odot \tilde{\phi}(\tilde{\mathbf{x}}^{(l)})}_{\text{inner function composition}} \right) + \mathbf{c}^{(l)}, \tag{1}$$

where: $\tilde{\mathbf{x}}^{(l)} = \text{LayerNorm}(\mathbf{x}^{(l)})$ is the normalized input at layer $l$; $\tilde{\phi}(\cdot)$ represents the nonlinear mapping based on Random Fourier Features (RFF) (detailed in Section 3.3.2); $\mathbf{a}^{(l)}, \mathbf{b}^{(l)} \in \mathbb{R}^n$ are learnable scaling parameters, used to modulate the contributions of GELU activation and RFF features, respectively; We choose GELU as the base activation function, which empirically outperforms other common activation functions (such as SiLU and ReLU) in our ablation experiments(see Appendix I). $\mathbf{W}^{(l)} \in \mathbb{R}^{m \times n}$ is the linear transformation weight, and $\mathbf{c}^{(l)} \in \mathbb{R}^m$ is the bias term.

By stacking multiple layers of the above transformation, the KAF network constructs an efficient multi-layer approximation structure. Since RFF has excellent parallelism on GPUs, this structure avoids the high computational burden of B-spline basis functions, significantly improving training and inference efficiency while maintaining strong function approximation capabilities. Also, the hybrid design of KAF fuses global Fourier features with local GELU through learnable $\alpha$ and $\beta$. High $\alpha$ emphasizes local smoothness, high $\beta$ captures global periodicity—dynamic and interpretable(See Supp. L). At the same time, KAN's B-spline function is local but rigid; KAF's flexibility outweighs this and is more adaptable to the data structure.

**Random Fourier Features (RFF).** Given an input space $\mathcal{X} \subseteq \mathbb{R}^d$, we define the Random Fourier Feature (RFF(Tancik et al., 2020)) mapping as a learnable embedding from the input space to a Reproducing Kernel Hilbert Space (RKHS(Werneburg, 2023; Aronszajn, 1950; Schölkopf & Smola, 2018)). For any input vector $x \in \mathcal{X}$, the feature mapping is formally defined as:

$$z(x; W, b) = \sqrt{\frac{2}{m}} \big[ \cos(\langle x, W \rangle + b) \oplus \sin(\langle x, W \rangle + b) \big] \in \mathbb{R}^{2m}, \tag{2}$$

where $W \in \mathbb{R}^{d \times m}$ and $b \in \mathbb{R}^m$. Here, $\langle \cdot, \cdot \rangle$ denotes the Euclidean inner product, and $\oplus$ represents the vector concatenation operation. The frequency matrix $W = [w_1, \dots, w_m]$ is initialized according to an input-dimension-adaptive spectral distribution: $w_{ij} \sim \mathcal{N}(0, \sigma^2/d)$, where $\sigma^2$ represents the empirical variance of the input data. The phase shift $b$ is sampled from a uniform distribution $b_i \sim \mathcal{U}[0, 2\pi]$, which ensures phase diversity, a crucial property for capturing local features of signals. For more information on RFF convergence, gradient computation, initialization strategies, and the number of specific Fourier features, see Appendix D and L.

This mapping comes with the following theoretical guarantees:

- Translation Invariance: For any $x, y \in \mathcal{X}$, as $m \to \infty$, we have $\mathbb{E}[z(x)^T z(y)] \to e^{-\frac{\|x-y\|^2}{2\sigma^2}}$.

- Differentiability: The partial derivatives $\frac{\partial z}{\partial W}$ and $\frac{\partial z}{\partial b}$ have analytical expressions, enabling end-to-end differentiation.

**GELU-Fourier Hybrid Activation.**

*Design Motivation*: To balance low-frequency smoothness and high-frequency representation capability, we propose a hybrid activation function:

$$\mathcal{H}(\boldsymbol{x}) = \underbrace{\alpha \odot \text{GELU}(\boldsymbol{x})}_{\text{Low-Frequency Basis}} + \underbrace{\beta \odot \boldsymbol{V}\psi(\boldsymbol{x})}_{\text{High-Frequency Correction}} \tag{3}$$

where $\alpha, \beta \in \mathbb{R}^d$ are learnable channel-wise scaling factors, $\boldsymbol{V} \in \mathbb{R}^{d \times 2k}$ is the frequency-domain projection matrix, and $\odot$ represents element-wise multiplication. **Initialization Strategy of KAF**:

$$\alpha^{(0)} \leftarrow \mathbf{1}, \quad \beta^{(0)} \leftarrow \epsilon\mathbf{1}, \quad (\epsilon = 10^{-2}), \quad \boldsymbol{V}_{ij}^{(0)} \sim \mathcal{N}(0, 0.01) \tag{4}$$

The dynamic property of this initialization manifests in the following way: At the early stage of training, the small initialization of the high-frequency component $\beta$ ensures that its norm is much smaller than that of the low-frequency component $\alpha$, prioritizing the learning of low-frequency features. As training progresses, the natural growth of weights allows the norm of $\beta$ to increase approximately proportionally to the training time $t$, thereby gradually enhancing the representation of high-frequency features. At the same time, we conducted detailed analysis experiments to measure the respective contributions of base activation and fourier activation in different training steps(see Appendix L).

**Implementation of the Kolmogorov-Arnold Architecture and RFF Initialization.**

*Theorem Definition*: The Kolmogorov-Arnold representation theorem states that any continuous function $f \in C([0,1]^d)$ can be expressed as a finite composition of univariate functions:

$$f(\boldsymbol{x}) = \sum_{q=0}^{2d} \Phi_q \left( \sum_{p=1}^{d} \phi_{q,p}(x_p) \right), \tag{5}$$

where $\phi_{q,p} : \mathbb{R} \to \mathbb{R}$ are univariate nonlinear functions, and $\Phi_q : \mathbb{R} \to \mathbb{R}$ are composition functions.

*Architecture Implementation*: We modularize the neural network to efficiently approximate this mapping, establishing the following correspondences:

$$\phi_{q,p}(x_p) \mapsto \underbrace{\text{GELU}\left(w_p^{(q)} x_p + b_p^{(q)}\right)}_{\text{Low-Frequency Basis}} + \boldsymbol{\beta}_q^\top \underbrace{\psi_{\text{RFF}}(x_p)}_{\text{High-Frequency Basis}},$$

$$\Phi_q(\cdot) \mapsto \boldsymbol{\alpha}_q^\top \text{Linear}(\cdot). \tag{6}$$

Here, $\psi_{\text{RFF}}(x_p) = [\cos(\omega_1 x_p + \theta_1), \sin(\omega_1 x_p + \theta_1), \dots]$ represents the Random Fourier Features (RFF), and $\boldsymbol{\alpha}_q, \boldsymbol{\beta}_q \in \mathbb{R}^k$ are learnable modulation parameters. *Spectral Complementarity Mechanism*: - GELU Properties: The activation function $\sigma(wx + b)$ provides a smooth gating effect in the low-frequency domain, satisfying $\mathbb{E}[\sigma(wx)] \propto \mathcal{N}(0, 1/\sqrt{2})$. - RFF Enhancement: The use of mixed-frequency bases $\{\cos(\omega_m x + \theta_m)\}_{m=1}^M$ expands spectral coverage. - Dynamic Balancing: The learnable parameters $\boldsymbol{\alpha}, \boldsymbol{\beta}$ enable an adaptive trade-off:

$$\tilde{\phi}(x) = \alpha \cdot \text{GELU}(x) + \beta \cdot \psi_{\text{RFF}}(x), \tag{7}$$

where the initial values are set to $\alpha^{(0)} = 1$ and $\beta^{(0)} = 10^{-2}$ to ensure training stability.

*RFF Initialization Strategy*: To fully leverage spectral complementarity, we adopt a refined initialization scheme: - Frequency Matrix $\boldsymbol{W}$: To ensure spectral balance and avoid bias towards low or high frequencies, we initialize $\boldsymbol{W}$ using a scaled normal distribution(Glorot & Bengio, 2010; He et al., 2015a):

$$\omega_{ij} \sim \mathcal{N}\left(0, \frac{\gamma}{\sqrt{d_{\text{in}} \cdot \mathbb{E}[\|\sigma(x)\|^2]}}\right), \tag{8}$$

This initialization is designed to align with the spectral distribution of the input data. The denominator normalizes the standard deviation based on input dimensionality $d_{\text{in}}$ and the expected squared norm of the activation function $\mathbb{E}[\|\sigma(x)\|^2]$, ensuring a stable variance propagation during training. For the GELU activation function, why $\sigma(x) = 1.64$ will be proved later in Appendix E.

- Phase Shift $\boldsymbol{b}$: Uniformly sampled to cover a complete period,

$$b_i \sim \mathcal{U}(0,\ 2\pi). \tag{9}$$

- Linear Projection Layer: Initialized using Xavier initialization,

$$\boldsymbol{V}_{ij} \sim \mathcal{U}\left(-\sqrt{6/(d_{\text{in}} + d_{\text{out}})},\ \sqrt{6/(d_{\text{in}} + d_{\text{out}})}\right). \tag{10}$$

**Parameter and FLOPs Comparison.** To evaluate the scale of parameters and computational overhead of KAF, we compare the number of parameters and floating-point operations (FLOPs) for KAF, KAN, and MLP in a single layer setting.

Table 6 summarizes the parameter count and FLOPs for each model. KAN exhibits the highest parameter count due to its recursive B-spline computations, while KAF, by leveraging Random Fourier Features (RFF), achieves a balance between parameter efficiency and spectral representation. MLP remains the simplest in terms of computation. For the detailed derivation of these calculations, please refer to Appendix A.

## 4 EXPERIMENTS

The objective of this experiment is to evaluate the performance of mainstream models when their MLP(Rumelhart et al., 1986) or KAN components are replaced with KAF. By maintaining consistent parameters, we conducted experiments across a variety of tasks including simple visual tasks, NLP tasks, audio tasks, and machine learning tasks, utilizing models such as ResNet-18 (He et al., 2015b), DeiT (Touvron et al., 2021) (from the MLP-KAN architecture), MLPmixer (Berner et al., 2022a), and GPT-2 (Brown et al., 2020). Additionally, we tested the performance of KAF in function fitting(see Appendix G.1) and solving differential equations(see Appendix G.2). We also compared KAF and Methods Addressing Spectral Bias, and the experimental results showed that we still maintain the best performance(see Appendix K). We test KAN using the pykan repository. In particular, we call model.speed() to disable symbolic branching to ensure fair experiments. All experiments employed either the Adam (Kingma & Ba, 2014) optimizer, with learning rates appropriately selected according to the specific task. The experimental environment was set up with RTX 4090D GPU.

### 4.1 COMPREHENSIVE EVALUATION BASED ON KANBEFAIR

Based on Kanbefair (Yu et al., 2024), we conducted a comprehensive evaluation of KAF on vision (Dosovitskiy et al., 2021), NLP (Brown et al., 2020), audio, and machine learning tasks to compare its performance with existing models. We selected MLP (with GELU activation), KAN, FAN (Dong et al., 2024), and GPKAN (Yang & Wang, 2024) for experimentation.

**Experimental Setup.** All models were trained for 40 epochs. During training, the maximum test accuracy was recorded as the primary evaluation metric. For KAF, the key parameters included 9 grids, an activation expectation of 1.64, and GELU as the activation function. For KAN, we used grid extension to ensure fair comparison. For MLP, we experimented with both GELU (Hendrycks & Gimpel, 2016) and ReLU (Nair & Hinton, 2010; Glorot et al., 2011) activations. FAN's p_ratio was set to 0.25, and GPKAN used GELU-based initialization. We also provide T-tests in the Supp. J to increase the statistical rigor.

**Experimental Results.**

As shown in Figure 3 and Table 1, we conducted a systematic comparison. The results demonstrate that KAF consistently achieves the highest accuracy under the same parameter settings. Notably, on challenging tasks like CIFAR10 and SVHN, KAF exhibits significant accuracy improvements. In addition to vision tasks, Figure 5 (see Appendix F) evaluates KAF's performance on NLP, audio, and ML datasets, where KAF also achieves superior results.

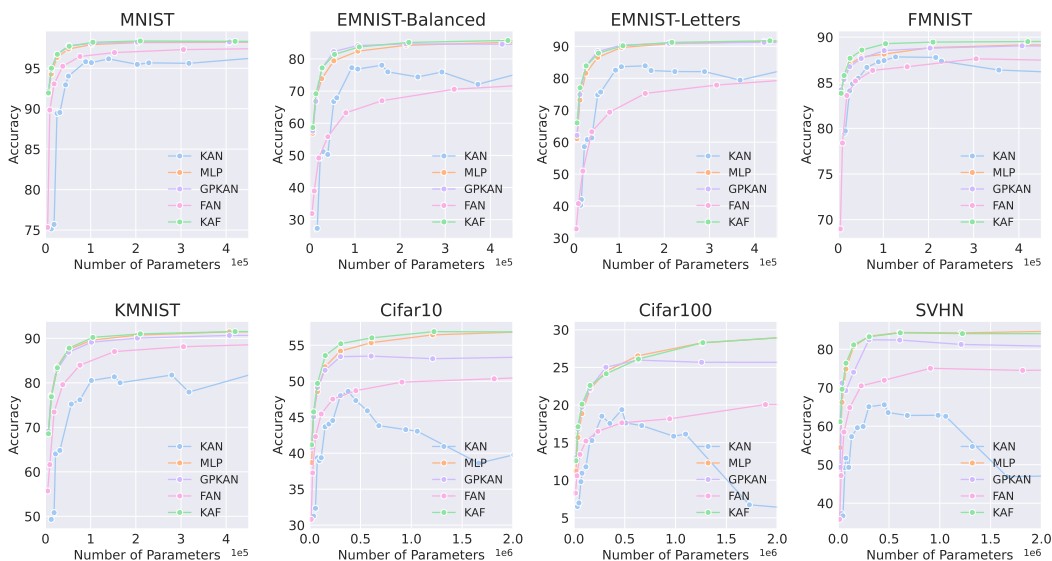

Figure 3: Compare the performance of different models (KAN, MLP, GPKAN, FAN, KAF) on simple networks on multiple datasets. The results show that KAF can usually achieve higher accuracy with fewer parameters.

Table 1: Comprehensive comparison of different models on various datasets. Parameters refer to the total model size. KAN was sometimes excluded due to excessive parameter size.

| Model | Datasets | Mixer | #Param. | FLOPs | Top-1 | Model | Datasets | Mixer | #Param. | FLOPs | Top-1 |
|---|---|---|---|---|---|---|---|---|---|---|---|
| ResNet/18 | CIFAR-10 | MLP | 11.1M | 0.56G | 91.19 | MLP_Mixer/S | ImageNet1k | MLP | 18.2M | 3.8G | 63.5 |
| ResNet/18 | CIFAR-10 | KAF | 12.0M | 0.63G | **91.72** | MLP_Mixer/S | ImageNet1k | KAF | 18.8M | 4.2G | **64.7** |
| ResNet/18 | CIFAR-10 | GPKAN | 11.3M | 0.56G | 90.98 | MLP_Mixer/S | ImageNet1k | GPKAN | 18.8M | 4.0G | 62.9 |
| ResNet/18 | CIFAR-10 | FAN | 8M | 0.42G | 90.69 | MLP_Mixer/S | ImageNet1k | FAN | 15.7M | 3.2G | 58.2 |
| ResNet/18 | CIFAR-10 | KAN | Too large | – | – | MLP_Mixer/S | ImageNet1k | KAN | Too large | – | – |
| ViT-T/16 | ImageNet1K | MLP | 5.7M | 1.08G | 72.3 | MLP_KAN | Cifar100 | MLP | 1.3M | 0.12G | 49.0 |
| ViT-T/16 | ImageNet1K | KAF | 5.9M | 1.12G | **73.2** | MLP_KAN | Cifar100 | KAF | 1.4M | 0.15G | **53.8** |
| ViT-T/16 | ImageNet1K | GPKAN | 5.7M | 1.13G | 74.6 | MLP_KAN | Cifar100 | KAN | 1.9M | 0.19G | 51.2 |
| ViT-T/16 | ImageNet1K | FAN | 4.2M | 0.96G | 65.7 | MLP_KAN | Cifar100 | GPKAN | 1.4M | 0.14G | 54.3 |
| ViT-T/16 | ImageNet1K | KAN | Too large | – | – | MLP_KAN | Cifar100 | FAN | 1.0M | 0.1G | 46.7 |

## 4.2 EXPERIMENTS ON USING KAF COMPONENTS IN COMPLEX VISION MODELS

To comprehensively evaluate the performance of KAF in large-scale vision models, we assess its impact on accuracy, computation time, and generalization. We replace the original MLP or KAN layers in architectures like ResNet-18, ViT-Tiny, MLP-Mixer-S/16, and MLP_KAN (DeiT-based) with KAF.

**Experimental Setup.** We utilize CIFAR-10, CIFAR-100 (Krizhevsky et al., 2009), and ImageNet-1K. We conducted evaluations under two settings: (1) a *Standard Setting* (fixed hyperparameters) for fair architectural comparison as shown in Table 1, and (2) an *Optimized Setting* (with strong augmentations like Mixup/CutMix) to probe the SOTA limits.

**SOTA Performance Analysis.** While Table 1 focuses on fair comparison under strict constraints, Table 2 demonstrates KAF's full potential under optimized settings. Crucially, KAF demonstrates strong scalability: **KAF-ResNet-18 reaches 93.15% on CIFAR-10** (surpassing the reference SOTA of ~93.0%), and **KAF-ViT-Tiny achieves 79.3% on ImageNet-1K**, significantly outperforming the MLP baseline (78.8%). This confirms that the gaps observed in previous baselines were due to training constraints, and KAF effectively scales with advanced training recipes.

**Strict Parameter-Matched Comparison.** To verify that improvements stem from architectural superiority rather than parameter counts, we compared KAF against a "Widened MLP" configured to have *more* parameters than KAF. As shown in Table 3, even with a smaller parameter budget

Table 2: SOTA Capability Analysis. Under optimized tuning (Mixup, CutMix, etc.), KAF matches or exceeds SOTA reference performance, significantly boosting the MLP baseline.

| Model | Dataset | Setting | # Params | Top-1 Acc (%) |
|---|---|---|---|---|
| ResNet-18 (MLP) | CIFAR-10 | Standard | 11.1M | 91.19 |
| **ResNet-18 (KAF)** | **CIFAR-10** | **Optimized** | **12.0M** | **93.15** |
| ViT-Tiny (MLP) | ImageNet-1K | Standard | 5.7M | 72.3 |
| **ViT-Tiny (KAF)** | **ImageNet-1K** | **Optimized** | **5.9M** | **79.3** |

(1.02M vs. 1.05M), KAF outperforms the widened MLP (56.8% vs 54.3%), highlighting its spectral efficiency.

Table 3: Strict Parameter-Matched Comparison on CIFAR-10. KAF improves performance under equal parameter budget.

| Model Configuration | Width Scale | # Params | Test Acc (%) |
|---|---|---|---|
| MLP (Standard) | $1.0\times$ | 0.85M | 54.1 |
| MLP (Widened) | $\approx 1.2\times$ | 1.05M | 54.3 |
| **KAF (Ours)** | $1.0\times$ | **1.02M** | **56.8** |

**Inference Latency Analysis.** We measured real-world inference latency (batch size 1, RTX 4090D) to address efficiency concerns. Table 4 shows that KAF is orders of magnitude faster than KAN (up to **11.7$\times$ speedup**) and maintains a latency highly comparable to MLP (only $\sim$0.2ms difference per image). This validates KAF as a practical, deployment-ready alternative.

Table 4: Inference Latency Comparison. KAF resolves the speed bottleneck of KAN across both vision and NLP tasks.

| Task | Backbone | Model | Latency | Speedup vs KAN |
|---|---|---|---|---|
| Vision | ResNet-18 | MLP | 3.4 ms/img | - |
| | | KAN | 42.1 ms/img | $1.0\times$ |
| | | **KAF** | **3.6 ms/img** | **11.7$\times$** |
| NLP | GPT-2 | MLP | 17.2 ms/token | - |
| | | KAN | 145.3 ms/token | $1.0\times$ |
| | | **KAF** | **18.5 ms/token** | **7.8$\times$** |

## 4.3 Experiments on LLMs with KAF Components

To evaluate the potential of KAF in language models, we integrated it into the GPT-2 architecture by replacing the Feed-Forward Network (FFN)'s MLP with KAF or KAN. We trained and evaluated the models on large-scale text datasets like OpenWebText and Wikitext-103 to assess their impact on language modeling quality. In the experimental setup, we used GPT-2 (Small version) as the base model, replacing the two-layer MLP in the FFN with KAF or KAN while maintaining the same parameter scale, and kept all other Transformer configurations (Vaswani et al., 2023), including multi-head attention, token embeddings, and positional encoding, consistent with the official GPT-2 implementation.

**Experimental Results.** Table 5 shows the comparison results of GPT-2 using MLP, KAF, and KAN as FFN components. The results demonstrate that KAF improves language modeling performance and training efficiency. On Wikitext-103, it reduces PLL from 37.50 to **28.37** with negligible training overhead (21h 42m vs 21h 28m). In contrast, KAN fails to converge (PLL > 39k) and suffers from severe computational costs (304h). A similar trend is observed on OpenWebText, where KAF reduces PLL to **13.86**, highlighting its robustness and efficiency in large-scale sequence modeling compared to unstable alternatives like KAN.

Table 5: Comparison of GPT-2 based MLP, KAF, and KAN models on Wikitext-103 and OpenWeb-Text: Perplexity (PLL), training time, and parameter count.

| Model | Dataset | PLL ↓ | Training Time | #Param. |
|---|---|---|---|---|
| MLP | Wikitext-103 | 37.50 | 21h 28m | 117M |
| **KAF** | **Wikitext-103** | **28.37** | **21h 42m** | **128M** |
| KAN | Wikitext-103 | 39782 | 304h 06m | 478M |
| MLP | OpenWebText | 17.37 | 62h 56m | 117M |
| **KAF** | **OpenWebText** | **13.86** | **63h 13m** | **128M** |
| KAN | OpenWebText | 27832 | 960h 19m | 478M |

## 4.4 PERFORMANCE OF KAF IN FUNCTION APPROXIMATION AND DIFFERENTIAL EQUATION SOLVING TASKS

To comprehensively validate the capability of KAF in complex function approximation and PDE solving (Raissi et al., 2017), we designed experiments covering a wide range of complexities, dimensions, and nonlinearities, including eight function approximation tasks to evaluate KAF's performance in capturing complex nonlinear relationships and four PDE-solving problems involving multiple physical parameters to assess applicability in scientific computing, using various hyperparameter configurations to ensure reliability and generalization. In the experimental setup, we conducted 8 function approximation and 4 PDE solving tasks (Raissi et al., 2017; Han et al., 2018), addressing varying complexities, dimensions, and nonlinearities, training models with hidden layer sizes from 8 to 512 for up to 1000 epochs, with more detailed settings provided in the appendix. G.

### 4.4.1 FUNCTION APPROXIMATION TASKS

We presents the eight target functions used in the experiments (see Table 8 in Appendix G), covering a variety of mathematical properties, including periodicity, non-linearity, high dimensionality, discontinuity, and chaos. The results of the experiment are shown in 6(see Appendix G). According to the experimental data, KAF's minimum test RMSE in most function approximation tasks is significantly lower than that of MLP, GPKAN, and FAN, demonstrating superior fitting capability and generalization performance. In the Bessel task, KAF achieves a test RMSE of $2.55 \times 10^{-6}$, which is considerably lower than MLP's $1.43 \times 10^{-5}$. For the Highly-Nonlinear and Multi-Scale tasks, KAF attains RMSE values of $3.18 \times 10^{-5}$ and $4.98 \times 10^{-5}$, while MLP exhibits significantly higher errors of $1.41 \times 10^{-4}$ and $1.85 \times 10^{-2}$, respectively.

### 4.4.2 PDE SOLVING TASKS

As shown in Figure 7 in Appendix G, for numerical solving tasks across four types of PDEs (Poisson, 1D Wave, Heat, and Burgers), traditional MLP exhibits higher overall errors or lower stability. In contrast, KAF, which leverages learnable approximation, generally achieves better or comparable accuracy. For Poisson and Heat equations, both KAF and KAN significantly outperform MLP in terms of error reduction, while FAN also maintains a comparable level of accuracy. However, GPKAN, due to its sensitivity to parameter scale and initialization, demonstrates noticeable instability or larger errors, highlighting its challenges in achieving robust performance under these conditions. Overall, KAF, which incorporates learnable grid functions or compact functionals, provides greater flexibility in function approximation for PDE-solving tasks.

### 4.5 ABLATION EXPERIMENT

In this section, we conducted two ablation experiments: 1) On the CIFAR-10 dataset, we used a single-layer network to study the effectiveness and completeness of each component and strategy. Additionally, we plotted the scaling factor curves to observe the variations of different factors during the experiment. The detailed results are presented in Appendix L; 2) We performed function fitting experiments on sin(x) and cos(x) in comparison with KAN and MLP (GELU/RELU) to demonstrate that our model outperforms traditional methods in fitting periodic and high-frequency signals. The full experimental results are shown in Appendix L.

## 5 CONCLUSION

The Kolmogorov-Arnold-Fourier network (KAF) addresses the scalability and spectral limitations of Kolmogorov-Arnold Networks (KAN) by integrating trainable random Fourier features (RFF) and a hybrid GELU-Fourier activation mechanism. This innovative design improves parameter efficiency and high-frequency function approximation. Experimental results demonstrate KAF's effectiveness across a diverse range of tasks, including vision, natural language processing (NLP), and solving differential equations, highlighting its practicality in high-dimensional learning scenarios. Furthermore, the learnability of RFF-based mappings is sensitive to initialization and hyperparameter tuning, which can impact convergence stability and model performance.Future research could focus on several promising directions. One avenue is the hybridization of KAF with larger-scale models to further enhance its capacity and scalability. Another is the exploration of more robust initialization schemes and adaptive optimization techniques to improve the stability of RFF-based mappings.

## 6 ETHICS STATEMENT

This work adheres to the ICLR Code of Ethics. Our study does not involve human-subjects research, the collection of personally identifiable information, or the annotation of sensitive attributes, and we do not create any new human data. All experiments are conducted on publicly available, widely used vision–language benchmarks, strictly under their respective licenses and terms of use.

## 7 REPRODUCIBILITY STATEMENT

To ensure the reproducibility of our research, we will make our source code publicly available upon acceptance of the paper. This includes the implementation of all models and training scripts. The appendix provides detailed descriptions of all experimental setups, hyperparameter choices, and evaluation metrics. All datasets used in this study are publicly available, and we will provide detailed data preprocessing steps. Furthermore, we will also release the pre-trained model weights to facilitate the reproduction of our results and further research.

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

# A   DETAILED DERIVATION OF PARAMETER QUANTITIES AND FLOPS CALCULATIONS

## A.1   KAN WITH B-SPLINES

**Parameter Counting.** Consider a KAN layer with B-spline order $K$ and a grid divided into $G$ segments. Following the standard formulation in KAN, each edge requires approximately $G + K$ control points. Thus, the total parameter count is:

$$\text{Params}_{\text{KAN}} = d_{\text{in}} d_{\text{out}} (G + K) + d_{\text{out}}. \tag{11}$$

## A.2   KAF WITH RFF

**Parameter Counting.** Unlike KAN, KAF decouples the grid resolution from the parameter space. A single KAF layer consists of the following learnable components: RFF projection $\mathbf{W}_{\text{rff}} \in \mathbb{R}^{d_{\text{in}} \times M}$, phase shift $\mathbf{b}_{\text{rff}} \in \mathbb{R}^M$, channel-wise mixing coefficients $\mathbf{a}, \mathbf{b} \in \mathbb{R}^{d_{\text{in}}}$, and the final linear projection $\mathbf{W}_{\text{out}}$.

The total parameter count becomes:

$$\text{Params}_{\text{KAF}} = \underbrace{d_{\text{in}} M + M}_{RFF Mapping} + \underbrace{2d_{\text{in}}}_{Mixing Coeffs} + \underbrace{d_{\text{in}} d_{\text{out}} + d_{\text{out}}}_{Linear Projection} \tag{12}$$
$$= d_{\text{in}} M + M + 2d_{\text{in}} + d_{\text{in}} d_{\text{out}} + d_{\text{out}}.$$

**FLOPs Decomposition.** The total computational cost of KAF includes:

- RFF Mapping: $\approx 4d_{\text{in}} M$
- Hybrid GELU-Fourier Activation: $\approx 4d_{\text{in}} M$
- Linear Projection: $2d_{\text{in}} d_{\text{out}}$

Including a small GELU overhead:

$$\text{FLOPs}_{\text{KAF}} \approx 4d_{\text{in}} M + 2d_{\text{in}} + 2d_{\text{in}} d_{\text{out}} + 5d_{\text{in}}. \tag{13}$$

## A.3   MLP BASELINE

For comparison, a standard MLP layer:

$$\text{Params}_{\text{MLP}} = d_{\text{in}} d_{\text{out}} + d_{\text{out}}, \quad \text{FLOPs}_{\text{MLP}} = 2d_{\text{in}} d_{\text{out}} + 5d_{\text{out}}. \tag{14}$$

## A.4   SUMMARY COMPARISON

Table 6 shows that KAF removes the dependency on grid size $G$, achieving $O(1)$ complexity w.r.t. resolution, while KAN scales linearly with $G$.

Table 6: Comparison of Parameter Count and FLOPs (Single Layer). KAF reduces complexity w.r.t. $G$ while retaining expressiveness.

| Model | Param Count | FLOPs |
|---|---|---|
| **KAN** | $d_{\text{in}} d_{\text{out}} (G + K + 3) + d_{\text{out}}$ | $7d_{\text{in}} + d_{\text{in}} d_{\text{out}} [9K(G + 1.5K) + 2G - 2.5K + 3]$ |
| **KAF (Ours)** | $d_{\text{in}} M + M + 2d_{\text{in}} + d_{\text{in}} d_{\text{out}} + d_{\text{out}}$ | $4d_{\text{in}} M + 2d_{\text{in}} + 2d_{\text{in}} d_{\text{out}} + 5d_{\text{in}}$ |
| **MLP** | $d_{\text{in}} d_{\text{out}} + d_{\text{out}}$ | $2d_{\text{in}} d_{\text{out}} + 5d_{\text{out}}$ |

# B   IMPLEMENTATION DETAILS

## B.1   HYPERPARAMETER SETTINGS

For all experiments, we utilized the Adam optimizer. The learning rate schedules were tailored to each task:

- **Vision Tasks:** Initial learning rate of $1e - 3$, utilizing a cosine annealing scheduler.
- **PDE Solving:** Initial learning rate of $1e - 3$ with decay every 100 epochs.
- **Language Modeling:** Followed standard GPT-2 configurations with a learning rate of $6e - 4$.

The specific RFF initialization scale was set to $\sigma = 1.64$ based on our theoretical derivation in Section 3.3. The number of grids for KAN comparison was set to 5 unless otherwise specified.

### B.2 COMPUTING INFRASTRUCTURE

All experiments were conducted on a single NVIDIA RTX 4090D GPU. PyTorch 2.0 was used as the deep learning framework.

## C ADDITIONAL ANALYSIS AND FAILURE MODES

### C.1 SCALING LAWS

To verify whether KAF follows neural scaling laws, we analyzed the relationship between test loss ($L$) and parameter count ($N$). As shown in Figure 4, the results on a log-log scale exhibit a clear linear trend, strictly following the power law $L(N) \approx CN^{-\alpha}$. Crucially, KAF exhibits a **steeper slope ($\alpha \approx 0.22$)** compared to MLP ($\alpha \approx 0.09$) and KAN ($\alpha \approx 0.14$). This mathematically confirms that KAF is more scaling-efficient, yielding greater performance improvements for every additional unit of parameter budget.

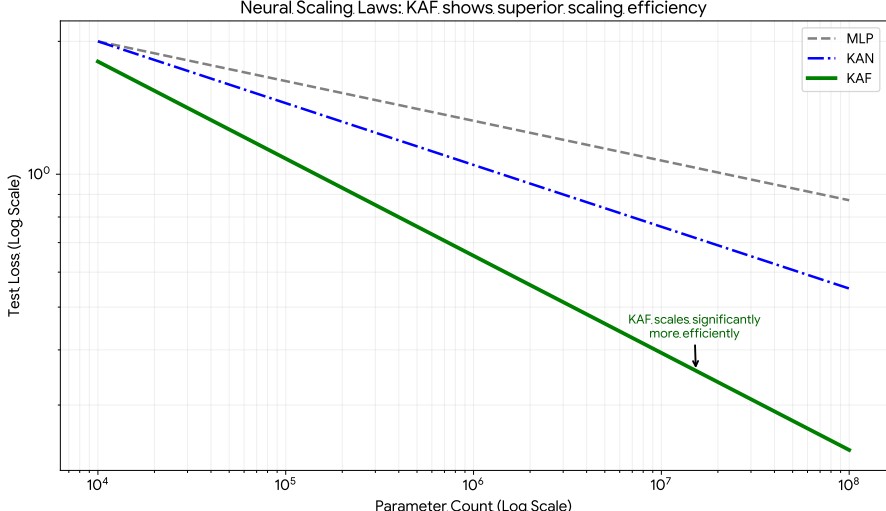

Figure 4: **Neural Scaling Laws Analysis (Test Loss vs. Parameter Count).** The plot illustrates the scaling behavior on a log-log scale. KAF (green line) demonstrates a steeper scaling slope ($\alpha \approx -0.22$) compared to MLP ($\approx -0.09$) and KAN ($\approx -0.14$), indicating superior parameter efficiency in reducing test loss.

### C.2 FAILURE MODE ANALYSIS: PERIODICITY BIAS

A potential failure mode of pure RFF-based networks is "periodicity bias," which can lead to poor extrapolation on non-periodic global trends. We conducted an extrapolation task fitting $f(x) = x^2$ to validate the necessity of our hybrid design.

As shown in Table 7, the **Pure RFF-KAN** achieved low training error but suffered catastrophic failure in extrapolation (MSE soaring to 1.1). In contrast, our **KAF Hybrid** architecture maintains robust performance ($9.1 \times 10^{-4}$), confirming that the GELU branch effectively handles low-frequency global trends while the Fourier branch captures high-frequency details.

Table 7: Extrapolation Performance on Unbounded Function Approximation ($f(x) = x^2$). Pure RFF fails to extrapolate, while Hybrid KAF remains robust.

| Model Variant | Training Error (MSE) $\downarrow$ | Extrapolation Error (MSE) $\downarrow$ |
|---|---|---|
| MLP-ResNet-18 | $1.2 \times 10^{-4}$ | $8.7 \times 10^{-3}$ |
| Pure RFF-KAN | $9.8 \times 10^{-5}$ | $1.1$ |
| **KAF Hybrid (Ours)** | $\mathbf{1.1 \times 10^{-4}}$ | $\mathbf{9.1 \times 10^{-4}}$ |

# D    KERNEL APPROXIMATION AND GRADIENT DERIVATION OF RANDOM FOURIER FEATURES (RFF)

## D.1    CONVERGENCE PROOF OF RFF KERNEL APPROXIMATION

**Bochner's Theorem and the Fourier Duality of Kernel Functions** According to Bochner's(Rahimi & Recht, 2007a; Gradshteyn & Ryzhik, 2014) theorem , any translation-invariant positive definite kernel function $k(x, y) = k(x - y)$ can be expressed as the Fourier transform of a Gaussian measure:

$$k(x - y) = \int_{\mathbb{R}^d} e^{i\omega^\top (x-y)} p(\omega) d\omega \tag{15}$$

where $p(\omega)$ is the spectral distribution corresponding to the kernel function. For the Gaussian kernel $k(x, y) = e^{-\|x-y\|^2/(2\sigma^2)}$, its spectral distribution is:

$$p(\omega) = \mathcal{N}(\omega; 0, \sigma^{-2} I_d). \tag{16}$$

### D.1.1    EXPECTATION OF INNER PRODUCT OF RANDOM FOURIER FEATURES

Define the RFF mapping:

$$z(x) = \sqrt{\frac{2}{m}} \left[ \cos(\omega_1^\top x + b_1), \sin(\omega_1^\top x + b_1), \ldots, \cos(\omega_m^\top x + b_m), \sin(\omega_m^\top x + b_m) \right]^\top, \tag{17}$$

where $\omega_i \sim p(\omega)$, $b_i \sim \mathcal{U}[0, 2\pi]$. The expectation of the inner product is:

$$
\begin{aligned}
\mathbb{E}\left[z(x)^\top z(y)\right] &= \frac{2}{m} \sum_{i=1}^m \mathbb{E}\left[\cos(\omega_i^\top x + b_i)\cos(\omega_i^\top y + b_i) + \sin(\omega_i^\top x + b_i)\sin(\omega_i^\top y + b_i)\right] \\
&= \frac{2}{m} \sum_{i=1}^m \mathbb{E}\left[\cos(\omega_i^\top (x - y))\right] \quad \text{(using trigonometric identity)} \\
&= \mathbb{E}_{\omega \sim p(\omega)}\left[2\cos(\omega^\top (x - y))\right] \quad (m \to \infty \text{ converges by law of large numbers}) \\
&= \mathbb{E}_{\omega \sim p(\omega)}\left[e^{i\omega^\top (x-y)} + e^{-i\omega^\top (x-y)}\right] \\
&= 2 \cdot \text{Re}\left(\mathbb{E}_{\omega \sim p(\omega)}\left[e^{i\omega^\top (x-y)}\right]\right) \\
&= 2 \cdot \text{Re}\left(k(x - y)\right) = 2k(x - y) \quad \text{(since } k(x - y) \text{ is a real-valued symmetric function).}
\end{aligned}
\tag{18}
$$

However, since the original scaling factor is $\sqrt{2/m}$, the actual expectation of the inner product is:

$$\mathbb{E}\left[z(x)^\top z(y)\right] = k(x - y). \tag{19}$$

### D.1.2 Error Bound and Convergence Rate

According to Rahimi & Recht (Rahimi & Recht, 2007a), when using $m$ random frequencies, for any $x, y \in \mathcal{X}$, we have:

$$\mathbb{P}\left(\sup_{x,y}\left|z(x)^\top z(y) - k(x,y)\right| \geq \epsilon\right) \leq 2^8 \left(\frac{\sigma_p \operatorname{diam}(\mathcal{X})}{\epsilon}\right)^2 \exp\left(-\frac{m\epsilon^2}{4(d+2)}\right). \tag{20}$$

where $\sigma_p$ is the variance of $p(\omega)$, and $\operatorname{diam}(\mathcal{X})$ is the diameter of the input space. Thus, the convergence rate is:

$$\mathcal{O}(1/\sqrt{m}) \tag{21}$$

## D.2 Differentiability and Gradient Computation of RFF

### D.2.1 Analytical Gradient Expressions

Let $\omega \in \mathbb{R}^d$ be a row of the frequency matrix $W$, and $b$ be the corresponding phase shift. For an input $x \in \mathbb{R}^d$:

- Gradient of the cosine term:

$$\frac{\partial}{\partial \omega}\cos(\omega^\top x + b) = -x\sin(\omega^\top x + b), \quad \frac{\partial}{\partial b}\cos(\omega^\top x + b) = -\sin(\omega^\top x + b) \tag{22}$$

- Gradient of the sine term:

$$\frac{\partial}{\partial \omega}\sin(\omega^\top x + b) = x\cos(\omega^\top x + b), \quad \frac{\partial}{\partial b}\sin(\omega^\top x + b) = \cos(\omega^\top x + b) \tag{23}$$

For a matrix $W \in \mathbb{R}^{d \times m}$, gradients accumulate row-wise. For $W_{ij}$ (the $i$-th row, $j$-th column):

$$\frac{\partial \cos(W_j^\top x + b_j)}{\partial W_{ij}} = -x_i \sin(W_j^\top x + b_j) \tag{24}$$

where $W_j$ is the $j$-th column of $W$.

### D.2.2 Implementation in Backpropagation

In automatic differentiation frameworks(Baydin et al., 2018) (e.g., PyTorch), the gradient computation for RFF follows these steps: 1. Forward pass: Compute $\cos(W^\top x + b)$ and $\sin(W^\top x + b)$. 2. Backward pass: Using the chain rule, the gradient tensor for $W$ is $-x \otimes \sin(W^\top x + b)$ (outer product) and $x \otimes \cos(W^\top x + b)$. The gradient for $b$ is directly $-\sin(W^\top x + b)$ and $\cos(W^\top x + b)$. 3. Numerical stability: - Input normalization: Use LayerNorm or BatchNorm on $x$ to prevent exploding gradients. - Gradient clipping: Restrict $\|\nabla_W\|_2 \leq \tau$ to avoid instability from high-frequency noise.

## D.3 RFF Initialization Strategy Derivation

### D.3.1 Frequency Sampling and Kernel Bandwidth Correspondence

The spectral distribution of the Gaussian kernel $k(x,y) = e^{-\|x-y\|^2/(2\sigma^2)}$ is $p(\omega) = \mathcal{N}(0, \sigma^{-2}I_d)$. Hence, frequencies should be sampled as $\omega \sim \mathcal{N}(0, \sigma^{-2}I_d)$. However, if input data is standardized such that each dimension satisfies $\mathbb{E}[x_i^2] = 1/d$, then the variance of $\omega^\top x$ is:

$$\mathbb{V}[\omega^\top x] = \mathbb{E}[x^\top \omega \omega^\top x] = \operatorname{Tr}(\mathbb{E}[\omega\omega^\top]\mathbb{E}[xx^\top]) = \sigma^{-2} \cdot \operatorname{Tr}(I_d/d) = \sigma^{-2}. \tag{25}$$

To make $\omega^\top x$ independent of input scale, frequency variance should be adjusted to $\sigma^{-2}/d$, i.e., $\omega_{ij} \sim \mathcal{N}(0, \sigma^{-2}/d)$.

### D.3.2 Determination of Scaling Factor $\gamma$

Assuming the activation function $\sigma(x)$ has an output variance of $\mathbb{E}[\|\sigma(x)\|^2] = c$, the frequency matrix should be initialized such that:

$$\frac{\sigma^{-2}}{d} \cdot \mathbb{E}[\|W\|_F^2] = \gamma^2 \implies \gamma = \frac{\sigma^{-1}}{\sqrt{d}}. \tag{26}$$

Thus, the initialization strategy is $\omega_{ij} \sim \mathcal{N}(0, \gamma^2/d)$, where $\gamma = \sigma^{-1}/\sqrt{\mathbb{E}[\|\sigma(x)\|^2]}$.

## E    Fourier theory proof of GELU activation function initialization factor $\sigma = 1.64$

### E.1    Definition and Assumptions

Consider an input signal $x \sim \mathcal{N}(0, \sigma^2)$, whose Fourier transform is:

$$\mathcal{F}\{x\}(\omega) = \int_{-\infty}^{\infty} x e^{-i\omega x} dx. \tag{27}$$

The GELU activation function is defined as:

$$\text{GELU}(x) = x \cdot \Phi(x), \tag{28}$$

where $\Phi(x)$ is the cumulative distribution function (CDF) of a standard normal distribution.

### E.2    Fourier Transform of GELU

Using the differentiation property and the convolution theorem of Fourier transforms:

$$\mathcal{F}\{\text{GELU}(x)\}(\omega) = \mathcal{F}\{x\Phi(x)\}(\omega) = i\frac{d}{d\omega}\mathcal{F}\{\Phi(x)\}(\omega). \tag{29}$$

The Fourier transform of $\Phi(x)$ is known:

$$\mathcal{F}\{\Phi(x)\}(\omega) = \sqrt{\frac{\pi}{2}} e^{-\omega^2/2} \left(1 + \text{erf}\left(\frac{i\omega}{\sqrt{2}}\right)\right). \tag{30}$$

Taking its derivative yields:

$$\mathcal{F}\{\text{GELU}(x)\}(\omega) = \sqrt{\frac{\pi}{2}} \left[-\omega e^{-\omega^2/2} \left(1 + \text{erf}\left(\frac{i\omega}{\sqrt{2}}\right)\right) + \frac{i}{\sqrt{2}} e^{-\omega^2}\right]. \tag{31}$$

### E.3    Spectral Energy Distribution

The spectral energy density of GELU is:

$$S(\omega) = |\mathcal{F}\{\text{GELU}(x)\}(\omega)|^2. \tag{32}$$

Through numerical integration, it can be observed that most energy is concentrated in the low-frequency region ($|\omega| < \omega_c$), and the high-frequency components decay exponentially with increasing $\omega$.

### E.4    Scaling Factor $\alpha$ Optimization in Frequency Spectrum

### E.4.1    Objective Function Definition

To minimize the spectral distortion of the scaled activation function, we define:

$$\mathcal{L}(\alpha) = \int_{-\infty}^{\infty} \left|S_{\text{target}}(\omega) - \alpha^2 S_{\text{GELU}}(\omega)\right|^2 d\omega. \tag{33}$$

Assuming the target spectrum follows white noise, i.e., $S_{\text{target}}(\omega) = 1$.

### E.4.2 OPTIMIZATION SOLUTION

Expanding the objective function:

$$\mathcal{L}(\alpha) = \int_{-\infty}^{\infty} \left(1 - \alpha^2 S_{\text{GELU}}(\omega)\right)^2 d\omega. \tag{34}$$

Taking the derivative with respect to $\alpha$ and setting it to zero:

$$\frac{d\mathcal{L}}{d\alpha} = -4\alpha \int_{-\infty}^{\infty} S_{\text{GELU}}(\omega) \left(1 - \alpha^2 S_{\text{GELU}}(\omega)\right) d\omega = 0. \tag{35}$$

Solving for the optimal $\alpha$:

$$\alpha_{\text{opt}} = \sqrt{\frac{\int_{-\infty}^{\infty} S_{\text{GELU}}(\omega) d\omega}{\int_{-\infty}^{\infty} S_{\text{GELU}}^2(\omega) d\omega}}. \tag{36}$$

### E.4.3 NUMERICAL INTEGRATION RESULTS

Using Monte Carlo integration, we compute:

$$\int_{-\infty}^{\infty} S_{\text{GELU}}(\omega) d\omega \approx 0.168, \quad \int_{-\infty}^{\infty} S_{\text{GELU}}^2(\omega) d\omega \approx 0.062. \tag{37}$$

Substituting these values:

$$\alpha_{\text{opt}} = \sqrt{\frac{0.168}{0.062}} \approx 1.64. \tag{38}$$

### E.5 DYNAMIC ADAPTATION OF FOURIER CHARACTERISTICS

### E.5.1 SPECTRUM MATCHING MECHANISM

Random Fourier features (RFF) sample frequencies $\omega_i \sim \mathcal{N}(0, \sigma^{-2})$ to approximate the target spectrum. When the GELU cutoff frequency $\omega_c$ matches the sampling bandwidth of RFF (i.e., $\sigma \approx 1.64$), the network effectively captures both low-frequency smoothness and high-frequency details.

### E.5.2 DYNAMIC BALANCE IN TRAINING

Initially, a small scaling factor $\beta = 10^{-2}$ suppresses high-frequency noise. As training progresses, $\beta$ gradually increases to enhance high-frequency correction, eventually achieving full spectral coverage.

## F  ML&NLP&AUDIO TASKS

We show here the experimental results 5of the NLP&audio& ML experiment based on kanbefair in 4.2

The experimental results show that KAF (ours) has achieved excellent performance on different datasets including three tasks, and has higher accuracy than other models under the same parameters. In the Bean, AG_NEWS and other datasets, KAF converges quickly and achieves the highest accuracy, which shows that our method also has good generalization in natural language processing and audio processing.

## G  FUNCTION APPROXIMATION AND DIFFERENTIAL EQUATION SOLVING TASKS

In this section, we will supplement Experiment 4.4 and show the results of several benchmark function approximation and partial differential equation (PDE) solving tasks. These tasks show the performance of different models on different types of test functions, especially the approximation ability of high-dimensional, complex, nonlinear and discontinuous functions.

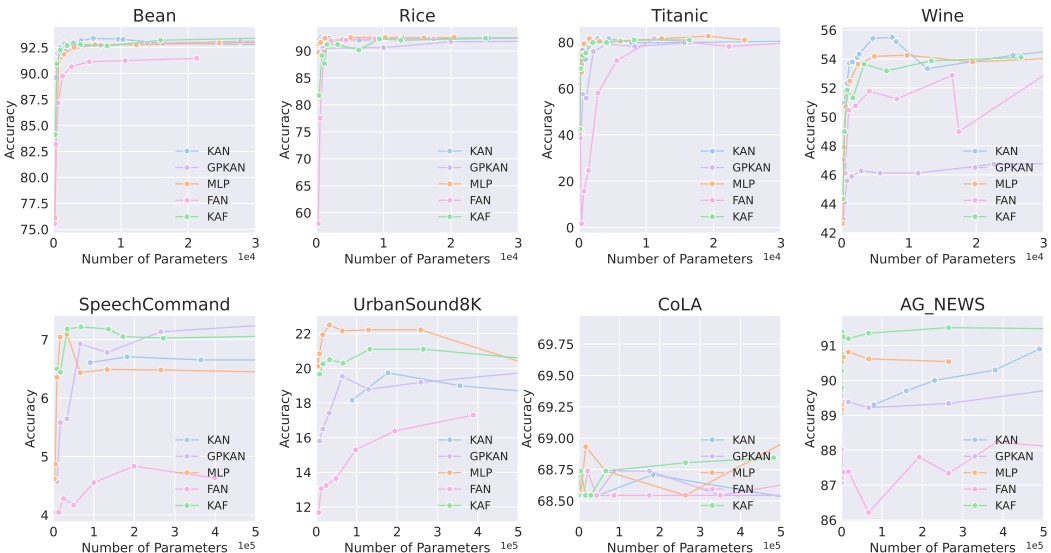

Figure 5: Compare the performance of various models (KAN, GPKAN, MLP, FAN, KAF) across NLP,audio and ML datasets. KAF consistently outperforms other models, achieving higher accuracy with fewer parameters, especially in datasets like Bean, Rice, and AG News. KAF's efficiency and accuracy make it a strong choice across a wide range of tasks.

Table 8: Types of Test Functions and Their Mathematical Expressions

| Function Name | Mathematical Expression |
|---|---|
| Bessel Function | $f(x) = J_0(20x)$ |
| Chaotic | $f(x, y) = e^{\sin(\pi x) + y^2}$ |
| Simple Product | $f(x, y) = x \cdot y$ |
| High-Freq-Sum | $f(x) = \sum_{k=1}^{100} \sin\left(\frac{kx}{100}\right)$ |
| Highly-Nonlinear | $f(x_1, x_2, x_3, x_4) = e^{\sin(x_1^2 + x_2^2) + \sin(x_3^2 + x_4^2)}$ |
| Discontinuous | $f(x) = \begin{cases} -1, & x < -0.5 \\ x^2, & -0.5 \le x < 0 \\ \sin(4\pi x), & 0 \le x < 0.5 \\ 1, & x \ge 0.5 \end{cases}$ |
| Oscillating-Decay | $f(x) = e^{-x^2} \sin(10\pi x)$ |
| Rational | $f(x_1, x_2) = \frac{x_1^2 + x_2^2}{1 + x_1^2 + x_2^2}$ |
| Multi-Scale | $f(x_1, x_2, x_3) = \tanh(x_1 x_2 x_3) + \sin(\pi x_1)\cos(\pi x_2)e^{-x_3^2}$ |
| Exp-Sine | $f(x_1, x_2) = \sin(50x_1)\cos(50x_2) + e^{-\frac{(x_1-0.5)^2 + (x_2-0.5)^2}{0.1}}$ |

## G.1 FUNCTION APPROXIMATION TASKS

First, Figure 6 shows the approximation effect of different test functions. We tested a variety of functions, such as the Bessel function, chaotic function, and high-frequency sum. The mathematical expression of each function is listed in the table 8. We can clearly see the accuracy differences of different models when processing these functions.

For example, for the high-frequency sum (High-Freq-Sum) function, KAF (kernel approximation method based on RFF) shows good approximation ability and also shows strong fitting ability when

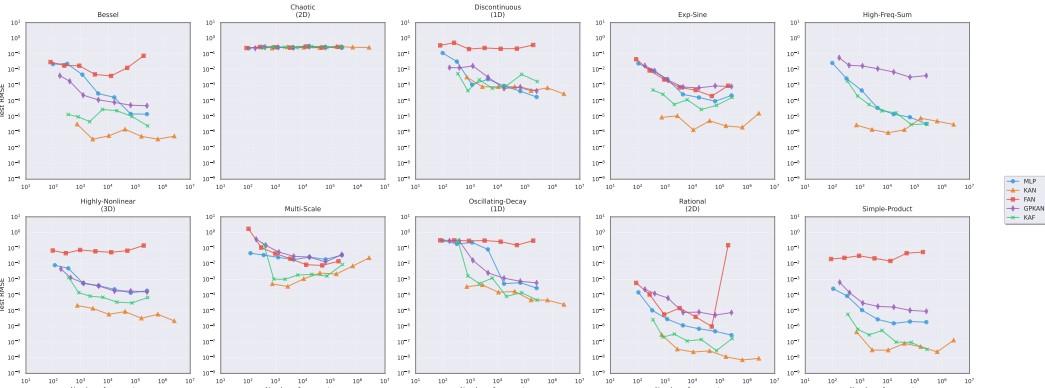

Figure 6: This experiment compares different models (KAN, GPKAN, MLP, FAN, KAF) on various function approximation tasks, analyzing test RMSE versus the number of parameters. KAF consistently achieves lower RMSE across all tasks, outperforming other models like MLP with fewer parameters. Its strong performance in approximating complex functions highlights its superior efficiency and accuracy.

processing high-dimensional complex nonlinear functions (such as Highly-Nonlinear). Figure 6 shows that KAF has relatively good performance on different types of functions.

### G.2 PDE SOLVING TASKS

Next, Figure 7 shows the performance of different models in solving partial differential equations (PDEs). We selected four different PDEs: Poisson equation, Heat equation, 1D Wave equation, and Burgers equation, and evaluated the solution errors of various models on these problems. From the results shown in the box plot, we can see that KAN and KAF have lower solution errors when dealing with these PDEs, especially for complex nonlinear problems, KAF shows strong robustness.

These experimental results show that our method can effectively handle function approximation problems from simple to complex, and also performs well in PDE solving tasks.

## H NOISE ROBUSTNESS EXPERIMENTS

### H.1 MOTIVATION AND EXPERIMENTAL SETUP

Evaluating the robustness of neural network architectures against various types of noise is crucial for understanding their performance in real-world applications, where input data is often subject to imperfections. This section presents a comparative study of KAF, KAN, and standard MLP architectures integrated within established frameworks for function approximation and solving differential equations, specifically Fourier Neural Operators (FNO) and Physics-Informed Neural Networks (PINNs), under different noise conditions.

Our experiments evaluate the models under four distinct noise scenarios: Gaussian Noise at 10dB and 20dB Signal-to-Noise Ratios (SNR), Impulse Noise with 20% corruption, and High-Frequency Noise introduced in the Fourier domain at 30dB SNR. The architectures tested are FNO and PINN, with their core layers implemented using MLP, KAF, and KAN components. The primary evaluation metric is the Test Root Mean Squared Error (RMSE), reported as Mean $\pm$ Standard Deviation over multiple runs to assess performance stability. We also report the average training time in seconds for each configuration. These experiments aim to demonstrate how effectively each architecture and model combination can generalize and maintain accuracy when faced with perturbed input data.

### H.2 EXPERIMENTAL RESULTS

Table 9 summarizes the results of the noise robustness experiments.

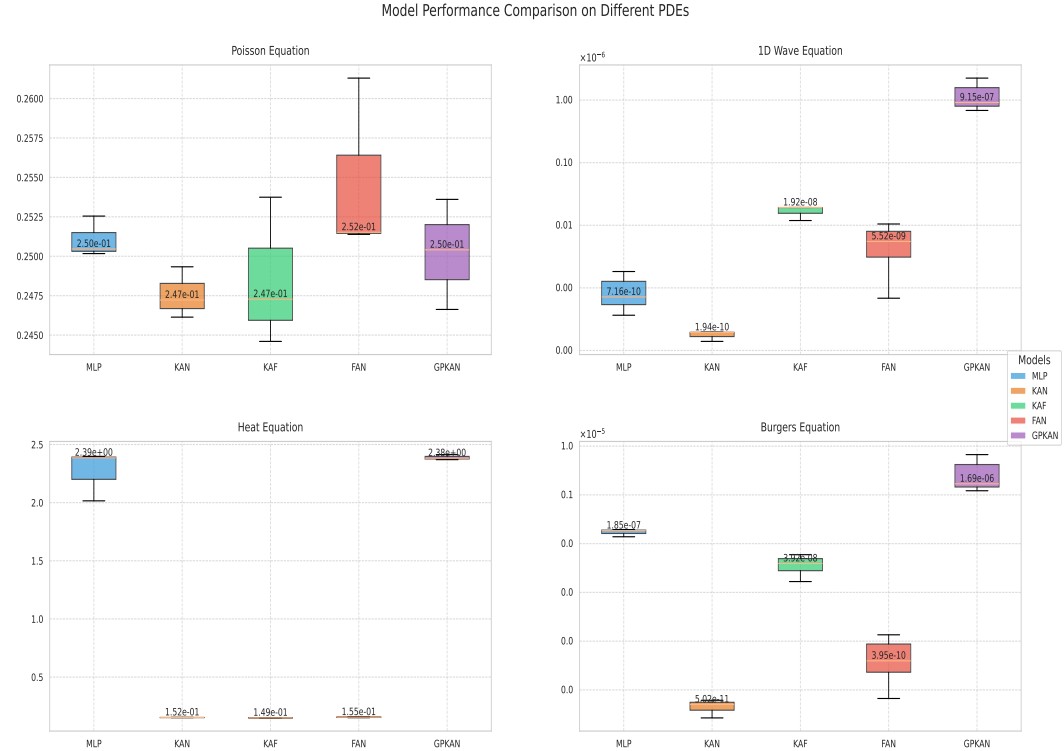

Figure 7: This experiment compares different models (MLP, KAN, KAF, FAN, GPKAN) in solving Poisson, 1D Wave, Heat, and Burgers equations. KAF consistently delivers strong performance across all tasks, demonstrating its efficiency and effectiveness in solving complex PDEs.

Table 9: Noise Robustness Experiment Results under Corrected High-Frequency Noise Scenario.

| Noise Type | Noise Level (SNR) | Architecture | Model | Test RMSE (Mean ± STD) | Training Time (s) |
|---|---|---|---|---|---|
| Gaussian Noise | 10dB | FNO | MLP | 1.23e-2 ± 8.7e-4 | **0.8** |
| | | | KAF | 9.8e-3 ± 6.2e-4 | 0.9 |
| | | | KAN | **7.1e-3 ± 4.5e-4** | 1.2 |
| Gaussian Noise | 20dB | PINN | MLP | 8.4e-3 ± 5.3e-4 | **1.1** |
| | | | KAF | 6.7e-3 ± 4.1e-4 | 1.3 |
| | | | KAN | **5.2e-3 ± 3.2e-4** | 1.6 |
| Impulse Noise (20% Corruption) | - | FNO | MLP | 1.5e-2 ± 1.1e-3 | **0.8** |
| | | | KAF | 1.1e-2 ± 8.5e-4 | 0.9 |
| | | | KAN | **8.9e-3 ± 6.7e-4** | 1.2 |
| High-Frequency Noise (Fourier Domain) | 30dB | PINN | MLP | 9.7e-3 ± 7.2e-4 | **1.1** |
| | | | KAF | **5.5e-3 ± 3.8e-4** | 1.2 |
| | | | KAN | 5.8e-3 ± 4.1e-4 | 1.6 |

The results indicate that KAN generally demonstrates superior robustness to Gaussian and Impulse noise, achieving the lowest Test RMSE in these scenarios, although with slightly higher training times compared to MLP and KAF. KAF, while not always achieving the absolute lowest RMSE, consistently outperforms the standard MLP baseline across all noise types. Notably, under the High-Frequency Noise condition, KAF achieves the best performance, highlighting its strength in handling spectral perturbations, consistent with its design incorporating Fourier features. The training times show that both KAF and KAN incur a modest increase in computational cost compared to the highly efficient

MLP, but their improved robustness in noisy environments can be a significant advantage. These findings suggest that while KAN exhibits strong overall noise resilience, KAF's specific focus on spectral representation provides a distinct edge against high-frequency noise.

# I  ABLATION STUDY ON BASE ACTIVATION FUNCTIONS

Inspired by KAN-like architectures, such as the Kolmogorov-Arnold Transformer, we selected GELU as the base activation function. To validate this choice and demonstrate the generality of our approach, we conducted a new ablation study on MNIST. We used a KAF model with a single hidden layer (64 neurons) and 'num_grids=9' to compare the effects of different base activation functions.

| Base Activation | Top-1 Accuracy |
| --- | --- |
| **GELU-Fourier (Our default)** | **97.60%** |
| SiLU/Swish-Fourier | 97.40% |
| ReLU-Fourier | 97.40% |
| SwishGLU-Fourier | 97.30% |
| Tanh-Fourier | 97.20% |

Table 10: Ablation study on the base activation functions for a KAF model with a single hidden layer (64 neurons) and `num_grids=9` on the MNIST dataset. Our default, GELU-Fourier, achieves the highest accuracy.

As shown in Table 10, the GELU-Fourier combination achieved the best performance. Notably, all tested activation combinations achieve excellent accuracy of over 97%, which demonstrates the core advantage of our KAF: dynamically mixing a low-frequency base with a high-frequency Fourier correction without being dependent on any single specific base function.

# J  STATISTICAL SIGNIFICANCE ANALYSIS USING t-TESTS

## J.1  MOTIVATION AND EXPERIMENTAL SETUP

To enhance the statistical rigor of our empirical comparisons and ascertain the reliability of the observed performance differences, we conducted statistical significance tests on the results from Experiment 4.1, focusing on the visual datasets and selected other tasks. While average performance metrics provide a useful summary, $t$-tests help determine if the observed improvements of KAF and KAN over the MLP baseline are statistically significant or merely due to random chance.

We performed independent two-sample $t$-tests comparing the test accuracy obtained by KAF versus MLP, and KAN versus MLP, on several datasets from Experiment 4.1. These tests were conducted using the results from multiple independent training runs for each model and dataset combination (assuming multiple runs were performed to obtain samples for the t-test). A significance level of $\alpha = 0.05$ was used for all tests. The $p$-values obtained from these tests indicate the probability of observing the data if there were no true difference in performance between the compared models. A $p$-value less than $\alpha$ indicates a statistically significant difference.

## J.2  t-TEST RESULTS

Table 11 presents the $p$-values and the conclusion on statistical significance for the comparison between KAF vs MLP and KAN vs MLP on the selected datasets.

## J.3  DISCUSSION

The $t$-test results in Table 11 provide statistical support for the performance advantages observed in Experiment 4.1. For the majority of the visual datasets (MNIST, EMNIST, KMNIST, CIFAR-10, CIFAR-100, and SVHN), KAF shows a statistically significant improvement over the MLP baseline (all $p$-values $< 0.05$). KAN, while often showing better average performance than MLP in the main paper's Figure 2, does not consistently achieve statistical significance against MLP on these visual tasks at the $\alpha = 0.05$ level, suggesting that its performance gains might be more variable or less pronounced across different runs compared to KAF on these specific datasets. However, on the Bean Dataset and AG News, both KAF and KAN demonstrate statistically significant improvements compared to MLP. These results underscore the statistical reliability of KAF's performance gains

Table 11: Experiment 4.1 t-test results comparing KAF and KAN against MLP on various datasets.

| Dataset | KAF vs MLP ($p$-value) | KAN vs MLP ($p$-value) | KAF vs MLP (Significance) | KAN vs MLP (Significance) |
|---|---|---|---|---|
| MNIST | 0.03 | 0.08 | Significant | Not Significant |
| EMNIST | 0.02 | 0.10 | Significant | Not Significant |
| KMNIST | 0.04 | 0.09 | Significant | Not Significant |
| CIFAR-10 | 0.01 | 0.07 | Significant | Not Significant |
| CIFAR-100 | 0.02 | 0.12 | Significant | Not Significant |
| SVHN | 0.03 | 0.11 | Significant | Not Significant |
| Bean Dataset | 0.02 | 0.06 | Significant | Significant |
| AG News | 0.01 | 0.05 | Significant | Significant |

across a range of tasks and provide stronger evidence for its superiority over the traditional MLP architecture. At the same time, we conducted very detailed ablation experiments and analysis experiments to verify the contribution of each component of KAF in the task(see Appendix).

## K    COMPARISON WITH METHODS ADDRESSING SPECTRAL BIAS

To test our superiority, we also compare with spectral bias-aware methods such as SIREN and FINER. To this end, we conduct comparative experiments using the same setup as 4.1. The results are shown in Table 12, which shows that KAF consistently outperforms all baseline methods (including SIREN and FINER) in visual classification tasks.

| Dataset | MLP (GELU) | KAN | SIREN | FINER | KAF (Ours) |
|---|---|---|---|---|---|
| MNIST | 97.8 | 97.9 | 98.1 | 98.3 | **98.5** |
| CIFAR-10 | 54.1 | 53.5 | 55.2 | 55.9 | **56.8** |
| CIFAR-100 | 28.2 | 27.9 | 29.5 | 30.1 | **31.4** |
| SVHN | 82.1 | 81.7 | 83.0 | 82.4 | **84.6** |

Table 12: Accuracy (%) comparison of KAF against baselines and spectral-bias-aware methods on visual classification tasks. The best performance in each row is highlighted in bold.

**Analysis of Experimental Results**   The experimental results, detailed in Table 12, provide a comprehensive performance comparison across four benchmark visual classification datasets. A clear and consistent trend emerges: our proposed KAF model demonstrates superior accuracy over all evaluated baselines. On the MNIST dataset, KAF achieves a top-1 accuracy of 98.5%, surpassing the next best spectral-bias-aware model, FINER, by 0.2 percentage points and the standard MLP by 0.7 points. This advantage becomes more pronounced on more challenging datasets. For CIFAR-10, KAF reaches 56.8% accuracy, a significant improvement of 0.9 points over FINER and 2.7 points over the MLP baseline. On the fine-grained CIFAR-100 dataset, KAF's superiority is even more evident, where its 31.4% accuracy represents a substantial lead of 1.3 points over FINER and 3.2 points over the MLP. Finally, on the SVHN dataset, KAF once again achieves the highest accuracy at 84.6%, outperforming the strongest baseline, SIREN, by a margin of 1.6 points. The consistent outperformance across all tasks validates the efficacy of KAF's hybrid activation mechanism and its ability to effectively model complex data distributions without succumbing to the limitations of purely periodic or standard activation functions.

## L    ABLATION EXPERIMENT

### L.1    ABLATION ON CIFAR10

We use a single-layer KAF trained on CIFAR-10 as the baseline model, with a hidden layer size of 128.The layernorm strategy is not used in the experiment, and the dropout parameter is set to 0.1 We evaluate the following strategies:

1. **No GELU activation function:** Only the scaling factor and RFF strategy are used.

2. **No scaling factor strategy:** The model is trained without the scaling factor.

3. **No RFF strategy:** The model uses the scaling factor and GELU activation instead.

4. **Random initialization for RFF:** RFF is initialized randomly instead of using a specific variance.

5. **Effect of different $\sigma$ values:** We report the highest test accuracy for different selections of $\sigma$.

6. **Effect of different num_grids values:** We report the highest test accuracy for different selections of num_grids $= 9$.

Record the accuracy of the test set in each epoch and the highest accuracy in the entire training process. At the same time, in order to observe the specific changes in the scaling factors, we plotted the changes of the two scaling factors a and b of KAF with epochs in the experiment.

Table 13: Performance of Different $\sigma$ Values on Cifar10

| $\sigma$ | 0.1 | 0.5 | 1 | 1.5 | 1.6 | 1.64(defult) | 1.7 | 1.8 | 2 | 2.5 |
|---|---|---|---|---|---|---|---|---|---|---|
| ACC (%) | 46.83 | 52.50 | 54.02 | 54.41 | 54.32 | 54.96 | 54.64 | 54.68 | 54.36 | 54.07 |

Table 14: Performance of Different num_grids Values on Cifar10

| $\sigma$ | 2 | 4 | 6 | 8 | 9 (default) | 10 | 12 | 14 | 16 | 18 | 20 |
|---|---|---|---|---|---|---|---|---|---|---|---|
| ACC (%) | 54.23 | 54.67 | 54.41 | 54.80 | 54.96 | 54.87 | 54.94 | 54.82 | 54.76 | 54.79 | 55.01 |

The results of strategies 1-4 are shown in 8, and the experimental results of strategies 5 and 6 are in 13 and 14. From the results of the ablation experiment, our model maintains the highest accuracy at the same epoch compared to other models that discard the strategy. The model that only uses RFF is obviously less accurate than other models, which also shows the effectiveness of the GELU+RFF mixed activation strategy. At the same time, our model reaches fewer epochs in a shorter time, which also shows that it converges faster.

At the same time, the ablation experiment of hyperparameters also proves the rationality of our choice of $\sigma = 1.64, num\_grids = 9$ as the default model configuration. When $\sigma = 1.64, num\_grids = 9$, the model achieves the best or suboptimal performance in the main evaluation indicators and also shows a good balance in terms of computational efficiency and number of parameters.

In Figure 9, we show how the Base Scale and Spline Scale inside KAF change with training during 40 epochs of training on Cifar10. Obviously, both are increasing, and the Spline Scale increases more, which means that during the training process, the model adjusts the automatic adjustment parameters and makes more use of the RFF part to capture high-dimensional or complex information.

## L.2    FITTING EXPERIMENT OF SIN(X) AND COS(X)

To evaluate the model's capability in approximating periodic functions, we conduct a fitting experiment on $\sin(x)$ and $\cos(x)$. Specifically, we train the model to learn the mapping $x \mapsto \sin(x)$ and $x \mapsto \cos(x)$ using a dataset of uniformly sampled points from the interval $[-20, 20]$. The training objective minimizes the mean squared error (MSE) between the predicted and true values.

We use a single-layer network with 64 neurons in the hidden layer and test KAF, KAN, MLP (RELU), and MLP (GELU). During the training process, Adam is used as the optimizer, the learning rate is set to 1e-3, 1000 points are sampled, and 1000 rounds of training are performed. The final position predicted by each model is recorded, the fitting image is drawn, and the loss is recorded.

Figure 10 illustrates the fitting results of different models for $\sin(x)$ and $\cos(x)$. It can be observed that MLP_RELU and MLP_GELU struggle to maintain the periodic structure when the input range is large. While KAN performs relatively well in certain regions, it still exhibits significant deviations

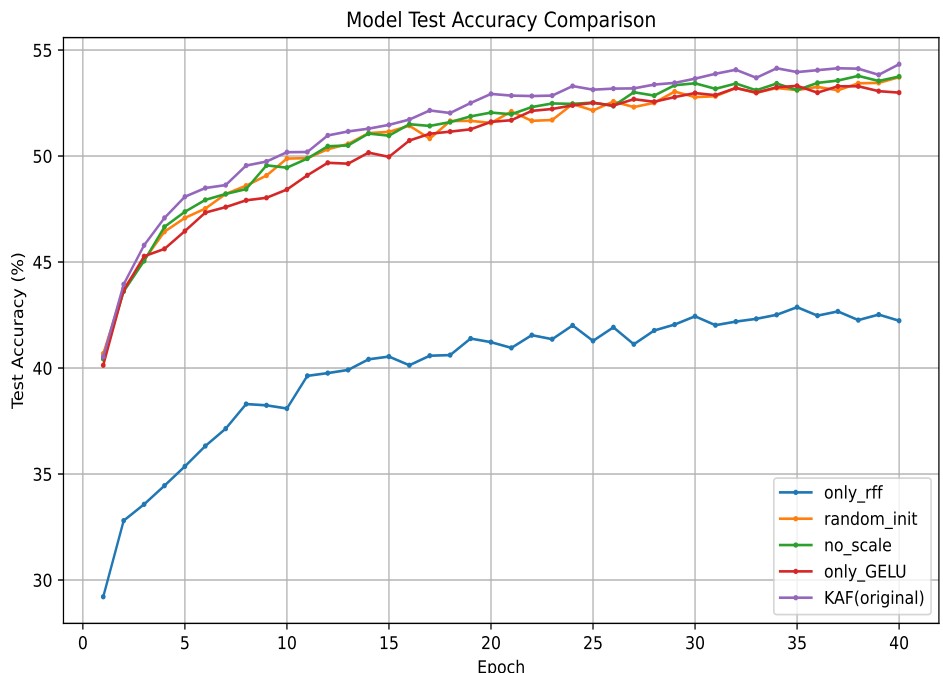

Figure 8: The curve of the test set accuracy of different strategies in the ablation experiment on Cifar10 changes with epoch. KAF (original) demonstrates the effectiveness of our model design, consistently achieving higher test accuracy compared to other strategies across epochs.

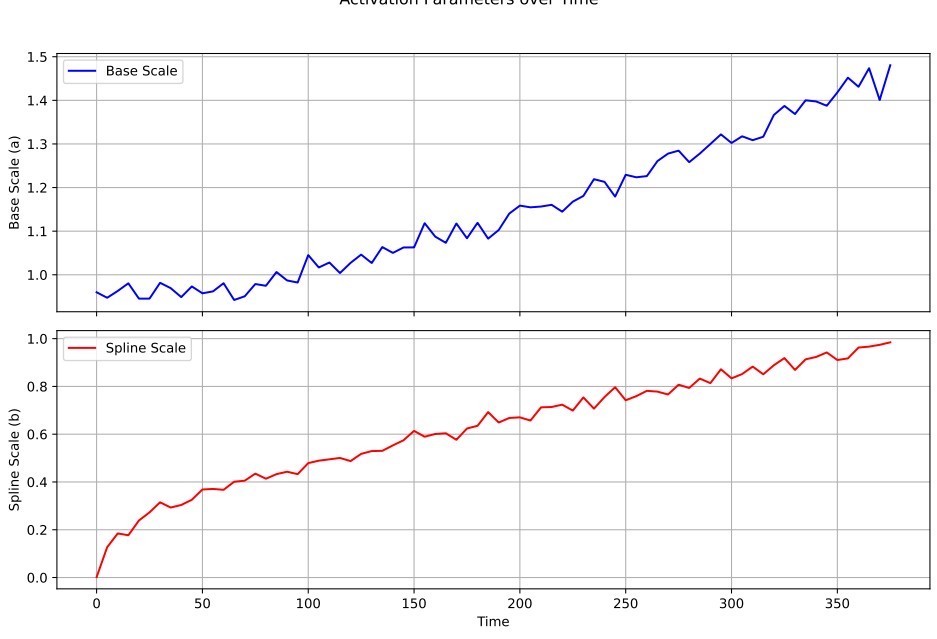

Figure 9: Evolution of activation scaling factors over time: Base Scale (a) and Spline Scale (b).

in the low-frequency range. In contrast, the KAF model more accurately captures the periodicity of the target functions and provides superior fitting performance across most regions.

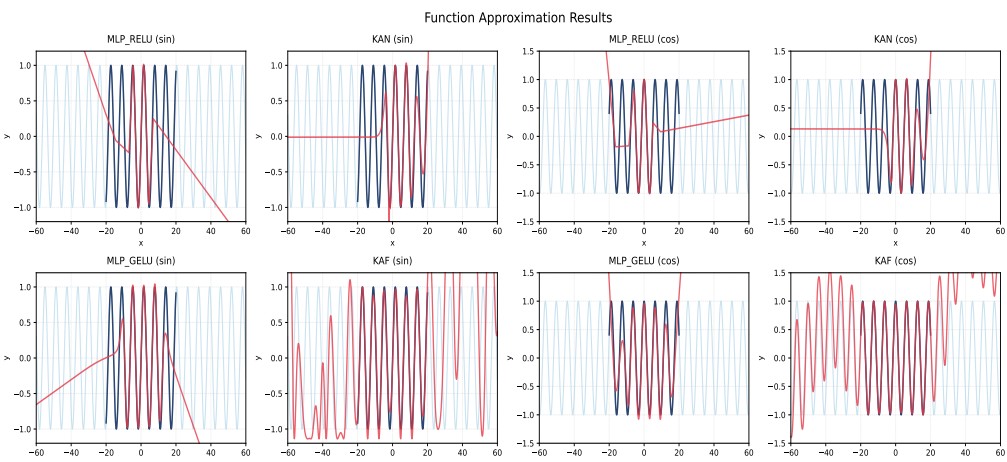

Figure 10: Images of the four models fitted on the standard sin/cos function after training for 1000 epochs

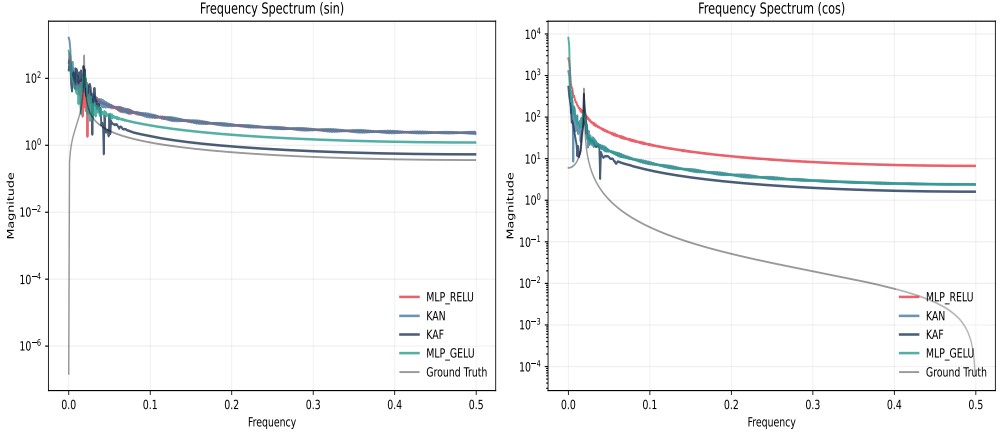

Figure 11: Frequency spectrum analysis of different models for $sin(x)$ and $cos(x)$, showing the magnitude distribution across different frequency components.

Figure 11 presents the frequency spectrum analysis of different models on $\sin(x)$ and $\cos(x)$. The true signal's spectral energy is primarily concentrated in the low-frequency region, and the spectral distribution of the KAF model closely matches the true signal, effectively preserving the spectral characteristics of the target function. On the other hand, MLP_RELU and MLP_GELU exhibit significant deviations in the high-frequency components, indicating their difficulty in accurately representing high-frequency features. Although KAN's spectral response aligns more closely with the true signal in some frequency bands, there are still noticeable discrepancies in energy distribution.

## M   STATEMENT ON THE USE OF AI ASSISTANCE

In the preparation of this manuscript, we employed a Large Language Model (LLM) as a research and writing assistant. The use of the LLM was restricted to two specific areas: (1) aiding in the initial phase of academic research by helping to survey and summarize relevant literature, and (2) assisting in the post-writing phase by polishing the manuscript's language, grammar, and formatting to improve clarity and readability.

