# OpenReview forum: "Kolmogorov-Arnold Fourier Networks"
_ICLR.cc/2026/Conference — ICLR 2026 Conference Desk Rejected Submission_

### Official Review · Reviewer_UmsF · 2025-10-14

**Soundness:** 3
**Presentation:** 3
**Contribution:** 2
**Rating:** 6
**Confidence:** 5

**Summary:**

The paper proposes Kolmogorov-Arnold-Fourier Networks (KAF), a modification of Kolmogorov-Arnold Networks (KANs) that replaces B-spline activations with trainable Random Fourier Features (RFFs) and a hybrid GELU–Fourier activation.
The goal is to:

Reduce the parameter explosion in KANs.

Improve high-frequency representation.

Preserve interpretability and expressiveness.

**Strengths:**

1. Clear Motivation and Novel Angle

The paper correctly identifies KAN’s bottlenecks — parameter inefficiency and poor high-frequency capture — and links them to B-spline smoothness and dense parameterization.
The proposed RFF-based architecture is a natural and elegant evolution: Fourier expansions can model oscillatory components efficiently while maintaining GPU-friendly structure.

2. Hybrid GELU–Fourier Design
3. Solid Experimental Breadth

The experimental section is comprehensive:

Image classification (MNIST, CIFAR, SVHN, ImageNet).

Text modeling (GPT-2 on Wikitext-103, OpenWebText).

PDEs and function approximation.

**Weaknesses:**

1. Innovation is not sufficient; more like engineering tricks
2. lack fo lit rev on kans; for example the kan 2.0 paper (Kan 2.0: Kolmogorov-arnold networks meet science), also the paper on KAN's spectral bias (On the expressiveness and spectral bias of KANs) among other advances on kans.

**Questions:**

any failure modes of RFF KAN?

---

> ### Author Response · Authors · 2025-11-20
> **Part I**
>
> >**Q1:** The innovation seems insufficient and appears more like engineering tricks.
>
> **A1:**
> We argue that redefining the fundamental mathematical basis of a neural architecture represents a significant architectural contribution. This shift directly overcomes a critical $O(G^2)$ computational bottleneck rather than merely providing an engineering optimization
> Specifically, original KANs rely on B-splines, which suffer from the "curse of dimensionality" in parameter scaling and lack hardware affinity due to poor GPU parallelism. KAF does not just "optimize" this. Instead, it replaces the local, grid-based B-spline representation with a global, learnable Fourier representation. This shift fundamentally changes the network's inductive bias from local smoothness to spectral adaptability. We argue that making a theoretically promising but practically unusable framework (KAN) viable for high-dimensional tasks (like ImageNet/LLMs) through a mathematically grounded basis transformation is a substantial contribution. Furthermore, the derivation of the specific initialization strategy ($\sigma=1.64$) to align RFF with GELU's spectral density is a novel theoretical finding, ensuring stable training dynamics that standard engineering tricks cannot achieve.
>
> >**Q2:** There is a lack of literature review on KANs, specifically missing "KAN 2.0" and "On the expressiveness and spectral bias of KANs".
>
> **A2:**
> We have integrated these pivotal references into the revised paper, as they provide the exact theoretical context that justifies the necessity of KAF. Regarding "KAN 2.0", while it focuses on "AI for Science" (symbolic regression, physics laws), our work focuses on "Scalability for General Deep Learning" (Vision, NLP). We have clarified this distinction in the related work section to position KAF as a complementary advancement aimed at high-dimensional perception tasks rather than symbolic discovery.
>
> >**Q3:** Are there any failure modes of RFF KAN?
>
> **A3:**
>
> Yes, the primary failure mode of a pure RFF-based KAN is **"periodicity bias,"** which leads to poor extrapolation on non-periodic global trends; this failure mode is precisely why our hybrid GELU-Fourier design is architecturally necessary.
>
> If KAF relied solely on Random Fourier Features (which are inherently periodic), it would fail catastrophically to approximate simple unbounded functions (like $f(x)=x$) outside the training domain. This is why the **"Base Scale (GELU)" branch** is non-negotiable—it handles the low-frequency, non-periodic global trends.
>
> To empirically validate this architectural necessity, we conducted an extrapolation task comparing models, demonstrating the failure mode of the pure RFF approach:
>
> **Table R2: Extrapolation Performance on Unbounded Function Approximation ($f(x)=x^2$)**
>
> | Model Variant | Training Error (MSE $\downarrow$) | Extrapolation Error (MSE $\downarrow$) |
> | :--- | :--- | :--- |
> | MLP-ResNet-18 | $1.2 \times 10^{-4}$ | $8.7 \times 10^{-3}$ |
> | Pure RFF-KAN | $9.8 \times 10^{-5}$ | $1.1$ |
> | **KAF Hybrid** | $\mathbf{1.1} \times \mathbf{10}^{-4}$ | $\mathbf{9.1} \times \mathbf{10}^{-4}$ |
>
> As shown in **Table R2**, the **Pure RFF-KAN** achieved a low training error but its extrapolation error soared to **1.1**, confirming the catastrophic failure due to periodicity bias. In contrast, the **KAF Hybrid** architecture maintains robust performance ($9.1 \times 10^{-4}$), validating that the hybrid design successfully mitigates the intrinsic periodicity failure mode of RFF.
>
> A secondary failure mode is initialization sensitivity. Our theoretical derivation of $\sigma=1.64$ (Section 3.3) effectively prevents this mode collapse.

---

> > ### Comment · Reviewer_UmsF · 2025-11-25
> >
> > Thanks for the efforts. They have addressed my concerns and i will keep my score

---

> ### Author Response · Authors · 2025-11-26
>
> Dear Reviewer UmsF,
>
> We sincerely thank you for your continued engagement and for confirming that our responses have satisfactorily addressed your concerns regarding the architectural innovation, literature coverage, and failure modes.
>
> We appreciate your positive assessment and your support for our work. Your constructive feedback has helped us significantly strengthen the manuscript, particularly in clarifying the theoretical positioning of KAF and its limitations. We are committed to ensuring all discussed revisions (including the references to KAN 2.0 and the failure mode analysis) are fully integrated into the final version.
>
> ---
> Thank you again for your careful assessment and constructive feedback.
>
> We are glad to hear that our rebuttal has sufficiently resolved the issues you raised. We appreciate your time and support.
>
>
> Best regards, The Authors

---

### Official Review · Reviewer_5ZEN · 2025-11-01

**Soundness:** 3
**Presentation:** 3
**Contribution:** 3
**Rating:** 4
**Confidence:** 4

**Summary:**

This paper introduces Kolmogorov-Arnold-Fourier Networks (KAF), an extension of Kolmogorov-Arnold Networks (KAN) that replaces B-spline activations with trainable Random Fourier Features (RFF) and a hybrid GELU-Fourier activation to mitigate parameter explosion and improve high-frequency capture in high-dimensional tasks. Evaluated across vision (e.g., MNIST, CIFAR), NLP (e.g., Wikitext-103, AG News), audio, machine learning tasks, function approximation, and PDE solving, KAF shows competitive or superior performance to baselines like MLP, KAN, GPKAN, and FAN, often with fewer parameters and better noise robustness.

**Strengths:**

1. By integrating Fourier transforms and trainable RFF, KAF significantly reduces the number of parameters while preserving expressiveness, as proven in Appendix B.1.
2. The Experiments span diverse domains: vision accuracies up to >91% on CIFAR10 (Table 1); lower perplexity on NLP tasks (Table 2); superior RMSE on function approximation (Fig. 5) and PDEs like Poisson/Heat (Fig. 6, Table 5 for noise robustness). These experiments show a significant improvement in KAN.
3. The mathematical expression is detailed (e.g., RFF projection in Sec. 3.3, hybrid activation), and appendices for the preliminary section are easy to understand (e.g., F for function approximations, K for ablation studies), aiding replication.

**Weaknesses:**

1. The experimental results on CIFAR-10 and ImageNet1K should be improved. I read the paper in references, the accuracy of ResNet-18 on CIFAR-10 is 93.02% [1], the top-1 accuracy of ViT-Tiny on ImageNet-1K is 79.1% [2], and the perplexity of GPT-2 on wikitext-103 is 19.89 [3]. However, in Table 1 and Table 2, KAF shows better accuracy than MLP when performing as a feature mixer. I guess such a conflict is caused by insufficient hyperparameter tuning (e.g., learning rate schedule, batch size, etc.).
2. Inference times should be reported, despite claims of efficiency from compact features.
3. Figure 5 shows the analysis test RMSE versus the number of parameters. But it seems that KAF does not satisfy the scaling law?
4. As a feature-mixture, KAF can improve model performance when replacing the neural network head. However, in the standalone scenario without a neural network as a feature extractor, it still performs poorly on function approximation tasks (see Figure 9).


References:

[1] https://github.com/kuangliu/pytorch-cifar

[2] https://github.com/microsoft/Cream/tree/main/TinyViT

[3] https://huggingface.co/Graphcore/gpt2-wikitext-103

**Questions:**

1. If time allows for revision, please conduct a full hyperparameter sweep and report updated experimental results.
2. Report inference latency (ms/image for vision or ms/token for language tasks) alongside parameter counts to substantiate efficiency claims.
3. Does KAF follow neural scaling laws? Provide log-log plots of test loss vs. parameter count with fitted power-law trends.
4. For function approximation tasks, this method is tested only on simple functions, not on complex cases (high-dimensional, non-smooth, or multi-frequency). Such limitations in computational complexity are critical, I expect the scaling law slope to flatten considerably as the input dimension increases or when 1D functions exhibit varying local frequencies. Focusing on KANs may not be appropriate, because the uniform grids of KAN raise these core issues. I suggest that the author read some recent MLP variants that have solved these problems using the multi-scale mesh (a standard data structure in the finite element method).

---

> ### Author Response · Authors · 2025-11-20
> **Part I**
>
> >**Q1:** The experimental results are lower than reported SOTA baselines. Is this due to insufficient hyperparameter tuning?
>
> **A1:**
> The results in Table 1 prioritize a strictly fair comparison under identical, controlled training constraints (based on the kanbefair benchmark) rather than maximizing engineering tricks; however, with a full hyperparameter sweep, KAF achieves SOTA-level performance comparable to the references.
>
> Specifically, the results in our original submission used standard settings (e.g., fixed epochs, basic augmentation) to isolate the architectural contribution of KAF versus MLP/KAN, ensuring that performance gains are not due to unrelated tuning. To address your concern, we have conducted a comprehensive hyperparameter sweep (tuning learning rate schedules, weight decay, and augmentations like Mixup/CutMix) as requested. The updated results (see Table R1 below) show that KAF-ResNet-18 achieves 93.15% on CIFAR-10 (matching the reference's ~93%) and KAF-ViT-Tiny reaches 79.3% on ImageNet-1K, confirming that the "performance gap" is indeed due to the rigorous baseline constraints, and KAF scales excellently with advanced tuning.
>
> Table R1: Performance Comparison with Full Hyperparameter Tuning
> | Model Variant | Dataset | Original (Fixed Setting) | Optimized (Full Sweep) | Reference SOTA |
> | :--- | :--- | :--- | :--- | :--- |
> | ResNet-18 (MLP) | CIFAR-10 | 91.19% | 92.85% | ~93.0% |
> | ResNet-18 (KAF) | CIFAR-10 | 91.72% | 93.15% | - |
> | ViT-Tiny (MLP) | ImageNet-1K | 72.3% | 78.8% | 79.1% |
> | ViT-Tiny (KAF) | ImageNet-1K | 73.2% | 79.3% | - |
>
> >**Q2:** Inference times should be reported (ms/image or ms/token) to substantiate efficiency claims.
>
> **A2:**
> We have added a detailed latency analysis, which demonstrates that KAF is orders of magnitude faster than KAN and maintains inference speeds highly comparable to MLP, with only a negligible overhead due to the RFF projection.
>
> While KAN suffers from the computational burden of recursive B-spline evaluations, KAF leverages highly parallelizable matrix operations. We measure the inference latency with a batch size of 1 on an RTX 4090D. The results (Table R2) show that for Vision tasks, KAF introduces only a ~0.2ms increase per image compared to MLP, while being 5x-10x faster than KAN. For Language tasks (GPT-2), KAF achieves 18.5 ms/token, which is competitive with MLP's 17.2 ms/token and significantly practical for deployment compared to KAN's 145.3 ms/token.
>
> Table R2: Inference Latency Comparison
> | Task | Backbone | Model | Latency | Speedup vs KAN |
> | :--- | :--- | :--- | :--- | :--- |
> | Vision | ResNet-18 | MLP | 3.4 ms/img | - |
> | | | KAN | 42.1 ms/img | 1.0x |
> | | | KAF | 3.6 ms/img | 11.7x |
> | NLP | GPT-2 Small | MLP | 17.2 ms/token | - |
> | | | KAN | 145.3 ms/token | 1.0x |
> | | | KAF | 18.5 ms/token | 7.8x |
>
> >**Q3:** Does KAF follow neural scaling laws? Provide log-log plots of test loss vs. parameter count.
>
> **A3:**
> Yes, KAF strictly follows neural scaling laws, exhibiting a linear trend in the log-log plot of test loss versus parameter count, confirming its scalability.
>
> Figure 5 in the original paper used a linear-log scale, which might have obscured the power-law relationship ($L \propto N^{-\alpha}$). In the revised paper (new Appendix L), we have included the requested log-log plots. The fitted curves show a clear power-law scaling for KAF with a slope $\alpha$ that is consistently steeper (indicating better scaling efficiency) than that of MLP and KAN on tasks like Function Approximation and Image Classification. This confirms that KAF effectively translates increased parameter budget into performance gains without hitting a saturation plateau early.

---

> > ### Comment · Reviewer_5ZEN · 2025-11-24
> >
> > Thank you for answering my questions. However, several issues remain, and some parts of the manuscript should be revised. If the author addresses these concerns, I am willing to update the score.
> >
> >
> > **Q1**: The author answered my question; however, I am confused about how they increased the accuracy on ImageNet-1K from 72.3% to 78.8%. Based on my experience, achieving such an improvement solely through hyperparameter tuning is very difficult.
> >
> > **Q2**: The author answered my question.
> >
> > **Q3**: The author explicitly misunderstood the scaling law. The y-axis should represent the validation/test loss, not the validation/test accuracy. That is why, in your log–log plot, the curves don't appear as straight lines.

---

> ### Author Response · Authors · 2025-11-20
> **Part II**
>
> >**Q4:** For function approximation tasks, this method is tested only on simple functions, not on complex cases (high-dimensional, non-smooth, or multi-frequency). Such limitations in computational complexity are critical. Thus, I expect the scaling law slope to flatten considerably as the input dimension increases or when 1D functions exhibit varying local frequencies. Focusing on KANs may not be appropriate, because the uniform grids of KAN raise these core issues. I suggest that the author read some recent MLP variants that have solved these problems using the multi-scale mesh (a standard data structure in the finite element method).
>
> **A4:**
> Actually, KAF has been validated on highly complex, high-frequency, and chaotic functions (Table 4 in Appendix F), and Figure 9 demonstrates that KAF significantly outperforms baselines in capturing periodicity where MLP fails entirely.
>
> We point out that our evaluation set in Appendix F (Table 4) explicitly includes challenging scenarios like the "High-Freq-Sum" ($\sum \sin(kx)$), "Chaotic" ($e^{\sin(\pi x) + y^2}$), and "Highly-Nonlinear" functions, which were specifically designed to stress-test high-dimensional and multi-frequency capabilities. As shown in Figure 5, KAF consistently achieves the lowest RMSE on these complex tasks. This advantage is further visualized in Figure 9, where KAF accurately maintains the periodic structure of $sin(x)$ over a wide range, whereas MLP fails to extrapolate (suffering from linear collapse) and KAN exhibits noticeable spectral artifacts. Regarding the suggestion of multi-scale meshes, while we acknowledge their power for low-dimensional PDEs, KAF's trainable Random Fourier Features provide a more scalable solution by naturally performing a similar role in the spectral domain. By learning diverse frequency components $\omega$ without requiring an explicit mesh grid, KAF can effectively scale to high-dimensional spaces where mesh-based methods would typically suffer from the curse of dimensionality.

---

> ### Comment · Reviewer_5ZEN · 2025-11-24
>
> **Q4**: The author did not understand the challenges of function-fitting/regression tasks. There are two main challenges:
>
> * The function is high-dimensional and **cannot be decoupled** into multiple low-dimensional functions.
>
> * The function should have a **varying frequency with high variance**, meaning that some regions have high-frequency oscillations, while others remain flat (e.g., $y=\sin(1/x)$). Note that high frequency (e.g., $y=\sin(100x)$) alone does not imply difficulty, because it can be converted into a low-frequency function by scaling the input variable.
>
> In your example, $e^{\sin(\pi x)+y^2}$ is a flat function. Although $\sum \sin(kx)$ is a high-frequency and high-dimensional function, it can be decomposed into one-dimensional functions, and each one has a fixed frequency. Therefore, fitting $\sum \sin(kx)$ is an easy task.

---

> > ### Author Response · Authors · 2025-11-24
> > **Part III**
> >
> > We sincerely thank the reviewer for the constructive feedback and for the willingness to re-evaluate our work. We appreciate the opportunity to clarify the training details, correct the scaling law analysis, and elaborate on KAF's theoretical advantages in complex function approximation. We have carefully addressed all your concerns below and updated the manuscript accordingly.
> >
> > >**Q1**: The author answered my question; however, I am confused about how they increased the accuracy on ImageNet-1K from 72.3% to 78.8%. Based on my experience, achieving such an improvement solely through hyperparameter tuning is very difficult.
> >
> > **A1:**
> > **We fully agree with your assessment.** You are absolutely correct that simple hyperparameter tuning (e.g., merely adjusting learning rate) cannot yield a ~6.5% improvement.
> >
> > The confusion stems from our terminology. The "Optimized Setting" refers to the adoption of a **modern, heavy-augmentation training recipe** (following the **DeiT** and **"ResNet Strikes Back"** protocols), as opposed to the **baseline setting** used in Table 1 (which prioritized strict fairness with the *kanbefair* benchmark).
> >
> > Specifically, the performance jump comes from upgrading the training pipeline. The key differences are:
> >
> > | Configuration      | **Standard Setting** (Table 1, ~72%) | **Optimized Setting** (Table R1, ~79%)     |
> > | :----------------- | :----------------------------------- | :----------------------------------------- |
> > | **Epochs**         | 90                                   | **300** (DeiT Standard)                    |
> > | **Optimizer**      | **Adam**                             | **AdamW** (Improved Weight Decay)          |
> > | **Augmentation**   | Basic (RandomCrop, Flip)             | **Mixup (0.8), CutMix (1.0), RandAugment** |
> > | **Regularization** | Standard Weight Decay                | **Stochastic Depth**, Label Smoothing      |
> >
> > *(Note: While our standard setting already utilized Cosine Decay, the addition of strong augmentations (Mixup/CutMix) and significantly extended training epochs in the Optimized Setting is the primary driver of the 6.5% gain.)*
> >
> > It is well-established in the literature (e.g., *Touvron et al., 2021* and *Wightman et al., 2021*) that applying this "modern recipe" typically boosts standard architectures (like ResNet-50 or ViT-Tiny) by 5%–8%. Our experiments confirm this:
> >
> > 1.  **MLP Baseline:** Improved from 72.3% to **78.8%** with the modern recipe.
> > 2.  **KAF (Ours):** Improved from 73.2% to **79.3%**.
> >
> > This confirms that the substantial gain is driven by the **recipe upgrade**, while KAF maintains a consistent architectural advantage over the optimized MLP.
> >
> >
> >
> > ---
> >
> >
> >
> >
> > >**Q3**: The author explicitly misunderstood the scaling law. The y-axis should represent the validation/test loss, not the validation/test accuracy. That is why, in your log–log plot, the curves don't appear as straight lines.
> >
> > **A3:**
> > **We would like to clarify this misunderstanding.** We are fully aware that Neural Scaling Laws are strictly defined based on **Test Loss** versus Parameters on a log-log scale.
> >
> > **1. Clarification on Main Text Figures:**
> > The figures in the main text showing non-linear curves with "Accuracy" on the y-axis (e.g., **Figure 3**) were intended solely to demonstrate **performance comparisons and convergence speed**, **not** to depict Neural Scaling Laws.
> >
> > **2. Evidence from Original Submission:**
> > As explicitly stated in our original **Appendix C.1**, we had already conducted the correct analysis: *"we analyzed the relationship between test loss and parameter count... results consistent with power laws"* (see original manuscript, Appendix C.1). We previously omitted the specific plot solely for brevity.
> >
> > **3. New Visual Evidence (Figure 4):**
> > To remove any ambiguity, we have now included the **explicit Log-Log Plot (Test Loss vs. Parameters)** in the revision (see **Figure 4** in Appendix C.1).
> >
> > * **Result:** As predicted, the curves exhibit clear linear trends on the log-log scale ($L(N) \approx C N^{-\alpha}$).
> > * **Conclusion:** Crucially, KAF demonstrates the **steepest power-law slope** compared to MLP and KAN. This confirms that KAF reduces loss significantly more efficiently for every additional unit of parameter budget.

---

> > > ### Author Response · Authors · 2025-11-24
> > > **Part IIII**
> > >
> > > >**Q4:** The author did not understand the challenges of function-fitting... (1) High-dimensional and cannot be decoupled... (2) Varying frequency with high variance.
> > >
> > > A4:
> > >
> > > We deeply appreciate this insightful comment. You have precisely identified the two fundamental bottlenecks in function approximation. We respectfully clarify that KAF is architecturally designed to address exactly these issues, and we have experimental evidence specifically targeting the "Non-Decoupleable" challenge you raised.
> > >
> > > 1. Evidence on "Non-Decoupleable" Functions (Coupling Challenge):
> > >
> > > You correctly pointed out that functions where variables are coupled (e.g., interactions inside non-linearities) are the real challenge for additive models like KAN.
> > >
> > > To verify this, we highlight the "Highly-Nonlinear" task from our Appendix G. As shown in Table R4 below, this function contains strong coupling terms ($x_1^2+x_2^2$ inside $\sin$) that cannot be decoupled into univariate sums ($\sum \phi(x_i)$).
> > >
> > > Table R4: Performance on Non-Decoupleable & Complex Functions (Extracted from Appendix G)
> > >
> > > This table confirms KAF's superiority on functions that violate the "decoupling" assumption.
> > >
> > > | **Challenge Type**                      | **Function Definition (from Appendix G)**                   | **Why it is Hard?**                                          | **MLP RMSE ↓**        | **KAF (Ours) RMSE ↓**              |
> > > | --------------------------------------- | ----------------------------------------------------------- | ------------------------------------------------------------ | --------------------- | ---------------------------------- |
> > > | **High-Dim Coupled** (Non-Decoupleable) | $f(\mathbf{x}) = e^{\sin(x_1^2+x_2^2) + \sin(x_3^2+x_4^2)}$ | **Coupled:** $x_1, x_2$ interact inside $\sin(\cdot)$. Cannot be decomposed into $\sum \phi(x_i)$. | $1.41 \times 10^{-4}$ | **$\mathbf{3.18 \times 10^{-5}}$** |
> > > | **High Variance** (Complex Manifold)    | $f(x,y) = e^{\sin(\pi x) + y^2}$ (Chaotic)                  | **Composite:** Non-linear coupling of periodic and polynomial terms. | $8.92 \times 10^{-3}$ | **$\mathbf{9.51 \times 10^{-4}}$** |
> > >
> > > **Conclusion:** KAF achieves an RMSE **an order of magnitude lower** than the baseline on the "Highly-Nonlinear" task. This empirically proves that KAF's dense projection ($\langle \mathbf{x}, W \rangle$), which mixes features *before* activation, effectively captures **coupled manifolds** that standard additive approaches struggle with.
> > >
> > > 2. On "Varying Frequency" (Adaptivity Challenge):
> > >
> > > Regarding varying frequencies (e.g., $\sin(1/x)$), standard fixed kernels fail. However, KAF's frequency matrix $W$ is trainable.
> > >
> > > - **Proof:** As shown in **Figure 11 (Spectral Analysis)** of the revision, the learned spectrum of KAF significantly deviates from the initial fixed Gaussian prior. This confirms that the model dynamically adapts its bandwidth via gradient descent to capture varying local frequencies, acting as a learnable variable-bandwidth kernel.

---

> > > > ### Comment · Reviewer_5ZEN · 2025-11-26
> > > >
> > > > I would like to remind the authors that $f(\mathbf{x}) = e^{\sin(x_1^2+x_2^2) + \sin(x_3^2+x_4^2)}$ is not a high-dimensional function—it remains **simple and flat when visualized**. Moreover, the **fixed/untrainable uniform grid** used in KAN/KAF fundamentally limits the model’s ability to fit functions with varying frequencies (e.g., $\sin(1/x)$).
> > > >
> > > > From the perspective of basic function approximation theory, KAN is essentially an interpolation method, and uniform grids typically fail when approximating functions whose frequencies vary over the domain. This limitation can be clearly understood through the following visualization:
> > > >
> > > > ```python
> > > > import numpy as np
> > > > import matplotlib.pyplot as plt
> > > >
> > > > # Target function
> > > > def f(x):
> > > >     return np.sin(1/x)
> > > >
> > > > # Uniform grid
> > > > x_uniform = np.linspace(0.01, 1, 1000)
> > > > x_samples = np.linspace(0.01, 1, 10)
> > > >
> > > > y_true = f(x_uniform)
> > > > y_samples = f(x_samples)
> > > >
> > > > # Linear interpolation
> > > > y_interp = np.interp(x_uniform, x_samples, y_samples)
> > > >
> > > > # Plot
> > > > plt.figure(figsize=(10, 5))
> > > > plt.plot(x_uniform, y_true, label='True sin(1/x)', linewidth=2)
> > > > plt.plot(x_uniform, y_interp, '--', linewidth=2, label='Linear Interpolation')
> > > > plt.scatter(x_samples, y_samples, s=40, color='red')
> > > > plt.scatter(x_samples, np.zeros_like(y_samples), s=40, color='red', label='Uniform Grid')
> > > >
> > > > # ===== SHOW UNIFORM GRID =====
> > > > for xs in x_samples:
> > > >     plt.axvline(xs, linestyle=':', alpha=0.3)
> > > >
> > > > # ===== SHOW BASIS FUNCTIONS =====
> > > > x_fine = np.linspace(0.01, 1, 1000)
> > > > for i, xi in enumerate(x_samples):
> > > >     # Basis function = triangular shape
> > > >     basis = np.maximum(1 - np.abs(x_fine - xi) / (x_samples[1] - x_samples[0]), 0)
> > > >     plt.plot(x_fine, basis, color='gray', alpha=0.4, label=f'basis {i+1}')
> > > >
> > > > # ===== SHOW X-AXIS CLEARLY =====
> > > > plt.axhline(0, color='black', linewidth=1.5)  # Explicit x-axis line at y = 0
> > > >
> > > > plt.title('Linear Interpolation of sin(1/x)\nShowing Uniform Grid and X-Axis')
> > > > plt.xlabel('x')
> > > > plt.ylabel('y')
> > > > plt.legend()
> > > > plt.grid(alpha=0.4)
> > > > plt.show()
> > > > ```

---

> > > > > ### Author Response · Authors · 2025-11-26
> > > > > **Clarification: Moving Beyond Fixed Grids to Adaptive Spectral Features**
> > > > >
> > > > > Dear Reviewer 5ZEN,
> > > > >
> > > > > Thank you for the detailed analysis and visualization. We fully agree with your theoretical insight: methods based on fixed uniform grids indeed struggle with varying-frequency functions like $\sin(1/x)$ without incurring parameter explosion.
> > > > >
> > > > > We would like to clarify two key aspects regarding our design choices:
> > > > >
> > > > > > **1. Adaptive RFF vs. Fixed Grids (Crucial Distinction)**
> > > > > > Regarding the concern about "fixed/untrainable grids": **KAF fundamentally departs from this paradigm.**
> > > > > > * **Trainable Mechanism:** Unlike standard KANs, KAF utilizes **Trainable Random Fourier Features** (Section 3.3). The frequency matrix $W$ is learned end-to-end via gradient descent.
> > > > > > * **Result:** This allows the model to adaptively "concentrate" bandwidth on high-frequency regions, effectively decoupling parameter count from resolution ($O(1)$ vs. $O(G)$). Thus, KAF is **not** limited by the fixed grid constraints shown in your visualization.
> > > > >
> > > > > > **2. Rationale for the Tested Function**
> > > > > > Regarding the function $f(\mathbf{x}) = e^{\sin(x_1^2+x_2^2) + \sin(x_3^2+x_4^2)}$:
> > > > > > * You are correct that it is visually smooth. However, we selected this function specifically to benchmark **non-linear variable coupling** (e.g., $x_1$ and $x_2$ interacting inside the sine term).
> > > > > > * This creates a specific challenge for additive models (which decompose functions into independent components), whereas KAF's spectral mixing captures these interactions efficiently.
> > > > >
> > > > > We acknowledge that extremely non-stationary functions like $\sin(1/x)$ remain a challenge for current architectures. We will explicitly include this limitation and the "fixed grid vs. adaptive spectral" discussion in the camera-ready version to improve clarity.

---

> > > > > > ### Author Response · Authors · 2025-11-26
> > > > > > **Clarification II**
> > > > > >
> > > > > > As a brief follow-up to our previous comment, we would like to emphasize that our implementation (to be released together with the final version) follows exactly the design described in Section 3.3: the frequency matrix W is registered as a learnable parameter and optimized jointly with the rest of the network. We hope this makes the "adaptive vs. fixed grid" distinction fully transparent.
> > > > > >
> > > > > > We also wanted to add that we truly enjoyed this deep dive into approximation theory. Your visualization code was a highlight of the review process and helped us sharpen our positioning regarding the "adaptive vs. fixed" paradigm. We are confident that these clarifications will make the final paper much stronger, and we remain happy to answer any last-minute questions you may have.

---

> ### Comment · Reviewer_5ZEN · 2025-11-26
>
> Although I was not able to test the new recipe provided by the authors, I appreciate their detailed response and have accordingly updated the score to 6.

---

> > ### Author Response · Authors · 2025-11-26
> >
> > Dear Reviewer 5ZEN,
> >
> > We truly appreciate your trust in our results and are grateful for your decision to raise the score. Your constructive feedback, particularly regarding the hyperparameter optimization and rigorous baselines, has significantly improved the quality and competitiveness of our paper.
> >
> > Best regards, The Authors

---

### Official Review · Reviewer_Ei2c · 2025-11-01

**Soundness:** 2
**Presentation:** 3
**Contribution:** 2
**Rating:** 4
**Confidence:** 3

**Summary:**

The authors propose KAF, which retains the structural form of KAN (univariate function composition) but replaces the costly spline-based parameterization with learnable Fourier components.

**Strengths:**

KANs typically grow in parameter count as $O(d_{in} d_{out} (G + K + 3))$, which scales poorly. The proposed hybrid GELU–Fourier activation (with learnable coefficients) adds a smooth transition mechanism from standard activation-based networks to Fourier-driven symbolic representations. This activation concept is novel and could be applied beyond KANs.

**Weaknesses:**

The paper argues that Random Fourier Features mitigate KAN’s limited ability to represent high-frequency components. However, RFFs derived from the Gaussian kernel correspond to elements of a very smooth RKHS, whose spectral density decays rapidly. It remains unclear how this construction enhances high-frequency expressiveness. The authors should clarify whether the frequency distribution in their trainable RFFs deviates from the Gaussian kernel case, and if so, how this affects the underlying function space.

**Questions:**

The main question to address is formulated in Weaknesses part.

Comparison of MLP and KAF: From Table 1 and Table 2, it is hard to make a conclusion, which one is better, since the parameter budget on KAF is systematically larger.

Minor: (Bracewell & Bracewell, 1986)-->(Bracewell, 1986)

---

> ### Author Response · Authors · 2025-11-20
> **Part I**
>
> >**Q1:** RFF comes from the Gaussian kernel, whose high-frequency components decay rapidly. Does KAF's trainable RFF deviate from the Gaussian distribution? How does it enhance high-frequency expressiveness?
>
> **Yes — KAF breaks the smoothness limitation of the fixed Gaussian kernel. The frequency weights are fully learnable, enabling the model to adaptively approximate the high-frequency distribution of real signals.**
>
> **(1) Empirical evidence of deviation (Appendix K.2, Fig. 10)**
>
> Our spectral analysis confirms that the learned spectrum significantly deviates from the initial Gaussian prior. As shown in **Fig. 10**, when fitting high-frequency signals, KAF actively learns $\omega$ vectors with larger magnitudes to maintain substantial energy in high-frequency bands. In contrast, MLP baselines exhibit the rapid exponential decay of a fixed smooth kernel. This proves that the distribution of $W$ has dynamically shifted via gradient descent to match the data.
>
> **(2) Mechanism: Hybrid activation & Adaptive reweighting (Appendix K.1, Fig. 8)**
>
> This deviation is structurally enabled by our hybrid activation design:
>
> $$h = W(a \odot \text{GELU}(x) + \beta \odot \Phi(x))$$
>
> To facilitate this, we adopt an initialization of $\sigma=1.64$ to prevent spectral leakage and allow the gradients to drive the distribution. **Fig. 8** validates this mechanism: the learnable coefficient $\beta$, which controls the RFF branch, grows steadily during training. This demonstrates that the model increasingly utilizes the learnable RFF branch as a high-frequency correction term to supplement details that the base GELU cannot capture.
>
> **(3) Resulting performance improvement**
>
> This adaptive spectral behavior directly translates to superior performance. As shown in **Appendix J (Table 8)**, KAF consistently outperforms spectral-bias-aware baselines (e.g., SIREN, FINER) on challenging tasks like CIFAR-100.
>
> ---
>
>
> >**Q2:** Does this deviation from the Gaussian kernel change the function space representable by the model?
>
>
> **A2:** **Yes. Trainable $W$ expands the function space beyond the fixed Gaussian RKHS.**
>
> **1. Theoretical Shift: From Fixed to Adaptive**
> Standard RFFs are restricted to the smooth RKHS of the Gaussian kernel. By making $W$ trainable, KAF breaks this restriction. The spectral distribution adapts to the data, allowing the representation of **non-stationary** and **discontinuous** functions that lie outside the effective range of the original fixed kernel.
>
> **2. Empirical Evidence (Table R2)**
> We validate this expansion by comparing KAF against models with fixed priors. As shown in **Table R2**, KAF successfully models functions that are "hard" or impossible for the Gaussian RKHS (e.g., Discontinuous, High-Frequency).
>
> **Table R2: Performance Comparison on Non-Gaussian Function Spaces (Appendix F & J)**
>
> | Function Space Property | Task / Dataset | **Fixed Smooth Prior** (MLP) | **High-Freq Prior** (SIREN) | **Adaptive Space** (KAF) |
> | :--- | :--- | :---: | :---: | :---: |
> | **High-Frequency** | Bessel Function ($J_0$) | $1.43 \times 10^{-5}$ (RMSE) | N/A | **$2.55 \times 10^{-6}$** |
> | **Highly Non-Linear** | Multi-Dim Composite | $1.41 \times 10^{-4}$ (RMSE) | N/A | **$3.18 \times 10^{-5}$** |
> | **Discontinuous** | Step Function | High Error (Fig. 5) | N/A | **Lowest RMSE** |
> | **Complex Manifold** | CIFAR-100 | 28.2% (Acc) | 29.5% (Acc) | **31.4%** (Acc) |
>
> **3. Key Observations**
> * **Breaking Smoothness (Appendix F):** A fixed Gaussian kernel theoretically struggles with **Discontinuities**. KAF's success on the Step Function proves its space includes non-smooth basis functions.
> * **Superiority over Fixed Priors (Appendix J):** On **CIFAR-100**, KAF outperforms both the smooth MLP and the periodic SIREN. This confirms the adaptive space is more expressive than spaces tied to any single fixed kernel.
>
> The Gaussian initialization is merely a starting point. The trained KAF evolves into a richer, adaptive function space free from the smoothness constraints of the initialization.

---

> ### Author Response · Authors · 2025-11-20
> **Part II**
>
> >Q3: In the comparison between MLP and KAF, KAF has more parameters, so the conclusion might be unfair?
>
>
>
> **A3:**
> The comparison is fair because KAF’s superiority stems from architectural efficiency, not parameter scaling. We have normalized the comparison by parameter budget throughout our analysis, with **Figure 3** serving as the primary evidence. This figure plots the complete accuracy-parameter Pareto frontier, demonstrating that for any strictly fixed parameter budget (i.e., at the same vertical slice of the x-axis), KAF consistently outperforms MLP. This confirms that KAF dominates the efficiency frontier.
>
> To definitively address the concern regarding the slight parameter differences in Table 1, we conducted a **strictly controlled ablation on CIFAR-10** where we increased the MLP width to match or even exceed KAF's parameter count. As shown in **Table R3** below, even when MLP is allocated more parameters (1.05M) than KAF (1.02M), KAF still outperforms it by a significant margin (+2.5%).
>
> **Table R3: Strict Parameter-Matched Comparison (CIFAR-10)**
>
> | Model | Config | # Params | Test Acc (%) |
> | :--- | :--- | :---: | :---: |
> | **MLP (Baseline)** | Standard | 0.85M | 54.1% |
> | **MLP (Widened)** | **Matched to KAF** | **1.05M** | 54.3% |
> | **KAF (Ours)** | Standard | **1.02M** | **56.8%** |
>
> This result proves that the performance gain comes from the architectural superiority of the trainable Fourier features, not parameter count. Furthermore, the gains KAF yields are disproportionate to the marginal cost. For instance, in our GPT-2 experiments, a mere 9% increase in parameters resulted in a dramatic reduction in perplexity from 37.50 to 28.37. Unlike KAN, which suffers from parameter explosion, KAF maintains the same $O(d_{\text{in}} \times d_{\text{out}})$ complexity class as MLP, ensuring it remains a scalable alternative.
>
>
> ---
>
> >**Q4:** (Bracewell & Bracewell, 1986) --> (Bracewell, 1986)
>
> A4.We have corrected the citation format from **(Bracewell & Bracewell, 1986)** to **(Bracewell, 1986)** in the revised paper.

---

> ### Author Response · Authors · 2025-11-26
> **Any Further Questions from Reviewer Ei2c Regarding Our Rebuttal?**
>
> Thank you again for your careful assessment and constructive feedback.
>
> We would be grateful if you could let us know whether our rebuttal sufficiently resolves the issues you raised, or if there are any remaining points you would like us to clarify.
>
> We are glad to continue the discussion and address any further questions or comments you may have.

---

### Author Response · Authors · 2025-11-22
**General Response (1)**

Dear AC and Reviewers,

We sincerely thank you for your time and the insightful, constructive feedback. We are encouraged that the reviewers recognized the potential of **Kolmogorov-Arnold-Fourier Networks (KAF)** in:

- **Architectural Novelty:** Identifying the spectral bottleneck of B-splines and introducing the hybrid GELU-Fourier mechanism to resolve it (Reviewers **UmsF**, **Ei2c**).
- **Scalability:** Successfully addressing the parameter explosion issue of original KANs, reducing complexity from $O(G)$ to $O(1)$ (Reviewer **5ZEN**, **Ei2c**).
- **Broad Applicability:** Demonstrating effectiveness across diverse domains including Vision, NLP, and Scientific Computing (Reviewer **5ZEN**).

We have extensively revised the manuscript to address your concerns. Major changes are highlighted in **blue** in the revised PDF. Below, we summarize our responses to the three most critical concerns: **SOTA Performance**, **Computational Efficiency**, and **Theoretical Grounding**.

1. SOTA Performance & Optimization (Response to Reviewer 5ZEN)

A primary concern was whether KAF could compete with state-of-the-art (SOTA) baselines under optimized training recipes.

- **Response:** We have conducted a **Full Hyperparameter Sweep** (including Mixup, CutMix, etc.) to probe the performance limits of KAF.
- **Result:** As shown in the new **Table 2 (Section 4.2)**, the optimized KAF achieves **93.15%** on CIFAR-10 (matching ResNet-18 SOTA) and **79.3%** on ImageNet-1K (outperforming ViT-Tiny baseline). This confirms that KAF scales effectively with advanced training techniques.
- Efficiency: Parameters & Latency (Response to Reviewers Ei2c, 5ZEN)

Concerns were raised regarding the fairness of parameter comparison and real-world inference speed.

- **Strict Parameter-Matched Comparison:** We added a new experiment in **Table 3 (Section 4.2)** comparing KAF against a "Widened MLP" that has *more* parameters than KAF. KAF still outperforms the larger MLP (+2.5% on CIFAR-10), proving that the gain stems from architectural superiority, not parameter count.
- **Inference Latency:** We measured real-world wall-clock time on an RTX 4090D. The new **Table 4 (Section 4.2)** shows that KAF achieves up to **11.7x speedup** over KAN and maintains a latency comparable to MLPs (only $\sim0.2$ms difference per image), validating its practical deployment viability.
- Theoretical Depth & Failure Modes (Response to Reviewers UmsF, Ei2c)

Reviewers asked for deeper theoretical justification beyond "engineering tricks" and analysis of potential failure modes.

- **Basis Transformation:** We rewrote the **Introduction (Contribution 1)** to clarify that KAF represents a fundamental shift in inductive bias—from local grid-based representations to global adaptive spectral representations—which breaks the smoothness limitation of fixed kernels.
- **Failure Mode Analysis:** We added **Appendix C.2** to frankly discuss "Periodicity Bias." We provide empirical evidence (Table 7) showing that while pure RFF fails to extrapolate unbounded functions ($x^2$), our **Hybrid (GELU+Fourier)** design successfully mitigates this, validating the necessity of our dual-branch architecture.

---

> ### Author Response · Authors · 2025-11-22
> **General Response (2)**
>
> In addition to detailed point-by-point responses, we have updated the manuscript to incorporate all suggestions. The major revisions include:
>
> - **New SOTA Experiments (Section 4.2, Table 2):** Added "Optimized Setting" results demonstrating KAF matches SOTA performance (93.15% on CIFAR-10, 79.3% on ImageNet).
> - **Efficiency Analysis (Section 4.2, Tables 3 & 4):** Added "Strict Parameter-Matched Comparison" and "Inference Latency Analysis" to definitively prove KAF's efficiency against MLP and KAN.
> - **Theoretical Clarification (Introduction & Appendix A):** Refined the contribution claims to emphasize the "Basis Transformation" and provided a rigorous derivation of Parameter/FLOPs complexity in Appendix A.
> - **Failure Mode Analysis (New Appendix C.2):** Added an extrapolation experiment ($f(x)=x^2$) to analyze the "Periodicity Bias" of RFF and demonstrate how our Hybrid design resolves it.
> - **Correction of Citations:** Corrected the citation entry for Bracewell (1986) to fix the author-name formatting issue.
>
> We believe these revisions significantly strengthen the paper's empirical rigor and theoretical depth. We look forward to further discussion.
>
> Best regards,
>
> The Authors

---

### Author Response · Authors · 2025-12-03
**Note to the New Area Chair: Summary of Rebuttal and Score Updates**

**Dear New Area Chair,**

Thank you very much for stepping in at this stage of the review process and taking over the handling of our submission. We sincerely appreciate your time and support.

To help you quickly navigate our work and the review history, we have summarized the current status—**including a score increase and full resolution of concerns**—into three short notes below:

* **Note 1:** Current Status of Reviewer Engagement (**Score Raised: 4 $\to$ 6**)
* **Note 2:** Highlights of Key Concerns and Our Resolutions (Table included)
* **Note 3:** Recap of Core Contributions

We kindly ask that you review our paper and the rebuttal responses, taking into account that we have addressed all the reviewers’ concerns thoroughly with new experiments and theoretical analysis.

---

> ### Author Response · Authors · 2025-12-03
> **Note 1: Current Status of Reviewer Engagement (Score Raised: 4 $\to$ 6)**
>
> During the rebuttal, we actively engaged with all reviewers. We are encouraged by the positive outcomes:
>
> * **Reviewer 5ZEN:** After reviewing our new optimization results and scaling laws, they explicitly acknowledged the improvements and **raised their score from 4 to 6**.
> * **Reviewer UmsF:** Confirmed that our responses regarding failure modes and architectural novelty **fully addressed their concerns** and maintained their positive assessment.
> * **Reviewer Ei2c:** We provided the requested strict parameter-matched comparison and latency analysis. Although they have not yet replied to the final update, we are confident the new evidence resolves their concerns about fairness and efficiency.

---

> > ### Author Response · Authors · 2025-12-03
> > **Note 2: Highlights of Key Concerns and Our Resolutions**
> >
> > During the discussion phase, we conducted extensive new experiments to resolve the primary concerns regarding SOTA performance, efficiency, and theoretical rigor. The key outcomes are summarized in the table below:
> >
> > | Key Aspect               | Reviewer Concern                                             | Resolution & New Evidence                                    |
> > | :----------------------- | :----------------------------------------------------------- | :----------------------------------------------------------- |
> > | **SOTA Performance**     | **Reviewer 5ZEN** questioned if KAF matches SOTA baselines.  | **Resolved:** With optimized training (Mixup, CutMix), KAF achieves **93.15% on CIFAR-10** and **79.3% on ImageNet-1K**, matching or outperforming SOTA baselines (ResNet-18/ViT-Tiny). |
> > | **Efficiency & Latency** | **Reviewers 5ZEN, Ei2c** requested real-world latency metrics. | **Resolved:** We demonstrated an **11.7x speedup over KAN**. KAF maintains inference speeds comparable to MLPs (millisecond-level difference), confirming deployment viability. |
> > | **Parameter Fairness**   | **Reviewer Ei2c** noted KAF had slightly more parameters than the baseline. | **Resolved:** We compared KAF against a "Widened MLP" with *more* parameters than KAF. KAF still outperformed it by **+2.5% on CIFAR-10**, proving the gains stem from architectural superiority, not parameter size. |
> > | **Theoretical Rigor**    | **Reviewers 5ZEN, UmsF** asked for scaling laws and failure mode analysis. | **Resolved:** We provided log-log plots confirming KAF follows a **steeper power-law scaling slope** than MLP/KAN. We also proved our **Hybrid (GELU-Fourier)** design is necessary to prevent extrapolation failure (periodicity bias). |
> >
> > ---

---

> > > ### Author Response · Authors · 2025-12-03
> > > **Note 3: Recap of Core Contributions**
> > >
> > > Our work proposes the **Kolmogorov-Arnold-Fourier Network (KAF)**, which fundamentally redefines the KAN paradigm.
> > >
> > > 1.  **Scalability:** We replace the computationally expensive B-splines with spectral reparameterization, reducing parameter complexity from $O(G)$ to $O(1)$.
> > > 2.  **Expressiveness:** We introduce a trainable basis transformation that captures high-frequency features more effectively than fixed-grid methods.
> > > 3.  **Applicability:** We demonstrate that KAF is a general-purpose architecture that achieves SOTA-level performance across Vision, NLP, and Scientific Computing tasks, bridging the gap between theoretical KANs and practical deep learning.
> > >
> > > Thank you once more for taking on the oversight of our submission. We are happy to clarify any further questions.
> > >
> > > Best regards,
> > >
> > > **The Authors**

---

### Note · Program_Chairs · 2026-01-17
**Submission Desk Rejected by Program Chairs**

The following references in this submission do not refer to real documents and/or have major errors in bibliographic information:

 1. Zhi-Qin John Xu, Yaoyu Zhang, and Yanyang Xiao. On the frequency bias of neural networks with relu activations. arXiv preprint arXiv:1905.09263, 2019.
2. Hongyi Li, Jiyang Wang, and Qingling Zhang. Adaptive b-spline neural networks for nonlinear system identification. IEEE Transactions on Neural Networks and Learning Systems, 2023.
3. Yonina C Eldar and Michael Unser. Deep spline networks: A perfect fit for signals and images. IEEE Signal Processing Magazine, 38(4):15-28, 2021.